# Twin-Merging: Dynamic Integration of Modular Expertise in Model Merging

**Zhenyi Lu**[1,2]* **Chenghao Fan**[1,2]* **Wei Wei**[1,2]† **Xiaoye Qu**[1] **Dangyang Chen**[3] **Yu Cheng**[4]

[1] School of Computer Science & Technology, Huazhong University of Science and Technology,
[2] Joint Laboratory of HUST and Pingan Property & Casualty Research (HPL),
[3] Ping An Property & Casualty Insurance Company of China, Ltd.,
[4] The Chinese University of Hong Kong.
{luzhenyi529,facicofan}@gmail.com, {weiw, xiaoye}@hust.edu.cn,
chendangyang273@pingan.com.cn, chengyu@cse.cuhk.edu.hk

## Abstract

In the era of large language models, model merging is a promising way to combine multiple task-specific models into a single multitask model without extra training. However, two challenges remain: (a) interference between different models and (b) heterogeneous data during testing. Traditional model merging methods often show significant performance gaps compared to fine-tuned models due to these issues. Additionally, a one-size-fits-all model lacks flexibility for diverse test data, leading to performance degradation. We show that both shared and exclusive task-specific knowledge are crucial for merging performance, but directly merging exclusive knowledge hinders overall performance. In view of this, we propose Twin-Merging, a method that encompasses two principal stages: (1) modularizing knowledge into shared and exclusive components, with compression to reduce redundancy and enhance efficiency; (2) dynamically merging shared and task-specific knowledge based on the input. This approach narrows the performance gap between merged and fine-tuned models and improves adaptability to heterogeneous data. Extensive experiments on 20 datasets for both language and vision tasks demonstrate the effectiveness of our method, showing an average improvement of 28.34% in absolute normalized score for discriminative tasks and even surpassing the fine-tuned upper bound on the generative tasks. [1]

## 1 Introduction

In recent years, Large Language Models (LLMs) have demonstrated notable success across various Natural Language Processing (NLP) tasks [12, 16, 43, 61–63, 65, 68], including code generation [22, 56], solving math problems [2, 44], multilingualism [47], *etc.* These models, with billions of parameters, excel in various downstream tasks [25, 34, 72] but require extensive training on large datasets using thousands of GPUs. The considerable computational and energy costs [53] limit their specialization and deployment in resource-constrained environments [38].

To tackle this challenge, model fusion has emerged as a promising solution [37]. One notable paradigm is model merging [29, 33, 76, 78], where multiple task-specific models, or "experts", are combined into a single unified model. This unified model can quickly adapt to new tasks without the need to retrain a large model. Various techniques, such as parameter averaging [6, 74], weight

---

* Equal contribution.
† Corresponding authors.
[1]Our implementation is available in https://github.com/LZY-the-boys/Twin-Merging

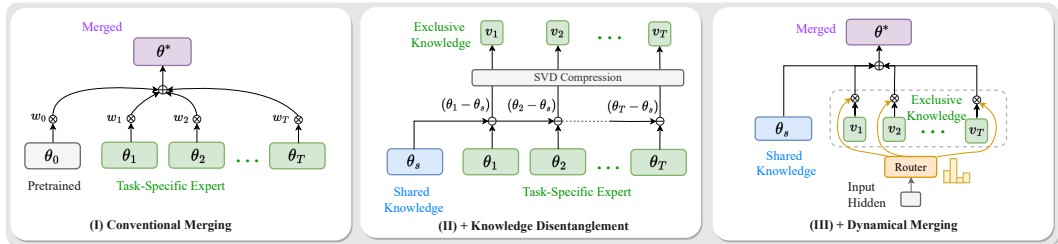

Figure 1: Subfigure (I) shows that in conventional merging methods, parameters from different task-specific models and a pre-trained model are weighted-summed into a single multitask model for inference. Subfigure (II) illustrates that our Twin-Merging method first isolates shared knowledge, then extracts exclusive knowledge by identifying differences between task experts and the shared model. This exclusive knowledge is then compressed into sparse vectors. Subfigure (III) shows that during testing, Twin-Merging dynamically merges shared and compressed specialized knowledge based on test inputs to form the final inference model.

interpolation [33, 46], and advanced strategies like task arithmetic [29, 51, 67, 78], have been developed for model merging. These techniques have been proven effective, enabling the integration of fine-tuned knowledge from diverse tasks into a multi-task model without additional training.

However, merging models from different domains often sacrifices specific task performance, leading to a large performance gap compared to the individual expert [31, 76]. Two major causes prevent the existing merging methods from reaching the theoretical upper-bound performance of individual experts: (1) *Interference between models.* Previous research shows that parameter redundancy and sign discrepancies [76], as well as the distribution gap between tasks [31], hinder effective model merging. We demonstrate that task-specific models often contain mixed knowledge, where the expertise in one model may be exclusive or detrimental to others. This redundancy or interference can obstruct the integration of expertise across models [9]. (2) *heterogeneity of data at test time.* Previous methods pursue a single, static optimal solution for various tasks. While a one-size-fits-all model avoids introducing new parameters, it might be inadequate or suboptimal due to the unpredictable nature of test inputs [78]. It limits the utilization of complementary knowledge and leads to deteriorated performance [71].

To address the above issues, in this paper, we introduce Twin Merging, involving two principal stages: (1) **Knowledge Modularization**: Unlike previous research that migrates merging interference in a parameter-wise manner or searches merging coefficients, we decompose the knowledge possessed by experts into shared knowledge and exclusive task-specific knowledge, as shown in Figure 1 (II). First, we compress common knowledge into a shared expert, serving to capture and consolidate common knowledge across varying tasks. Then we isolate exclusive knowledge based on the difference between the task experts and the shared expert, allowing diverse knowledge to be decomposed more finely. (2) **Dynamic Merging**: Inspired by Mixture of Experts (MoE) [80, 84, 85], we simplify the parameter merging problem into a conditional composition problem. Instead of pre-determining the best parameter combination for heterogeneous data at test time, as illustrated in Figure 1 (III), we introduce a router to dynamically merge shared and exclusive knowledge based on the test inputs. The shared model serves as the foundation, and task-specific knowledge is conditionally injected according to the router.

We demonstrate the effectiveness of our proposed Twin-Merging method through extensive experiments on 12 datasets, covering both discriminative and generative tasks, various model architectures, and in-domain and out-of-domain setups. As shown in Figure 2b, Twin-Merging consistently outperforms other merging methods across all datasets, surpassing the strongest baseline by an average of 28.34% in normalized scores for discriminative tasks and 3.86% for generative tasks on the scaled model (Qwen-14B). We validate the scalability, extensibility, generalization, and storage efficiency of Twin-Merging (Figure 2a). Remarkably, even with a 99.9% reduction in parameters, our method only experiences a slight 14% performance degradation. Our results establish Twin-Merging as a powerful and effective method for combining multiple fine-tuned models into a single multi-task model.

To summarize, our contributions are as follows: (1) We introduce Twin-Merging, a novel model fusion method that reduces the performance gap between traditional model merging and fine-tuned models while enhancing adaptability to diverse data. (2) We investigate the impact of shared and exclusive task-specific knowledge on merging performance, presenting innovative techniques for

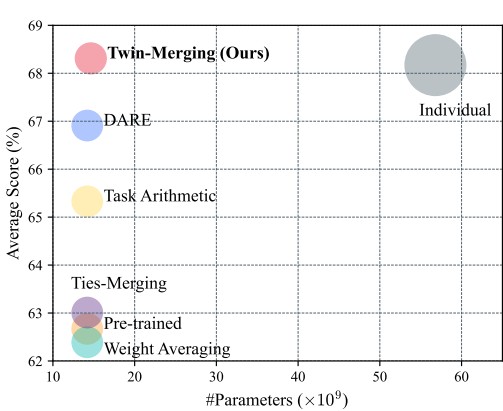 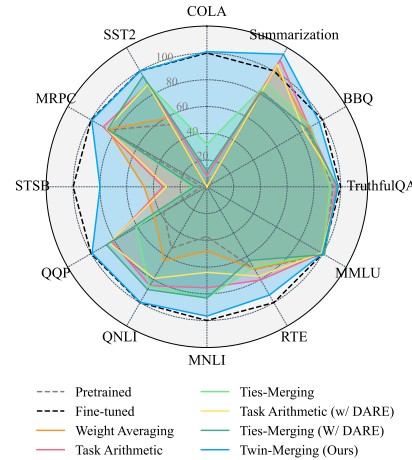

(a) The average performance on generative tasks *vs.* the number of parameters of Twin-Merging compared to various merging baselines, with different storage sizes indicated by circle size.

(b) Comparison of absolute accuracy (%) of individual tasks for the NLP benchmarks on RoBERTa and Qwen, covering 4 discriminative and 8 generative tasks.

Figure 2: The effectiveness of Twin-Merging in terms of performance and parameter-efficiency.

knowledge disentanglement and dynamic merging. (3) Twin-Merging is simple to implement with minimal hyperparameters, improves multi-task performance without retraining expert models, and can be combined with other merging methods for further gains. Our approach scales well with model size and task numbers and is storage-efficient.

## 2    Related Work

In this section, we focus on model merging research, for additional related work on multi-task learning and Mixture of Experts, please see Appendix B. Model merging aims to fuse multiple fine-tuned task-specific models into one comprehensive multi-task model without additional training. FisherMerging [46] and RegMean [33], use straightforward weight averaging but require extra data and computation. Some works [1, 21, 58, 60, 70] bring models into a single low-loss basin and interpolate between them based on the linear mode connectivity (LMC) theory [15, 18, 20]. The weight permutations [1] and optimal transport [58] are utilized to better interpolate neural networks. However, recent studies [83] suggest that LMC might not always hold for fine-tuned models. Task-Arithmetic [28, 51] extends averaging to arithmetic operations in the parameter space for finer control over model behaviors, but the interference between the multiple models can be an issue. To tackle this challenge, advanced merging methods like Ties-Merging [76], AdaMerging [78] and DARE [79] have been proposed. These methods aim to reduce task conflicts by addressing parameter redundancy or disagreements in signs, finding optimal merging coefficients, and reducing weight density, respectively. Jiang et al. [32] assume that test tasks are known and use task-specific knowledge to improve performance. However, this assumption is often unrealistic since real-world data distributions are unpredictable. In contrast, our method addresses merging interference by modularizing shared and task-specific knowledge. We handle heterogeneous test data scenarios by introducing dynamic merging techniques.

## 3    Methodology

### 3.1    Analysis of the Performance Gap in Model Merging

In this paper, following the settings of model merging [29, 76, 79], we consider the case of $T$ tasks, where training for each task $t$ starts from pre-trained model weight $\boldsymbol{\theta}_0$ and fine-tunes on $\mathcal{D}_t^{train}$ to obtain task-specific model $\boldsymbol{\theta}_t$. Let $f(\boldsymbol{x}; \boldsymbol{\theta})$ be a language model accepting inputs $\boldsymbol{x} \in \mathcal{X}$ and paramterized by weights $\boldsymbol{\theta} \in \Theta$. Considering the real data distributions are diverse and challenging to represent with a single task, to model such distributions, previous methods typically consider the mixture of $T$ task test data: $\mathcal{D} = \sum_{t=1}^{T} \alpha_t \mathcal{D}_t$, where $\sum_{t=1}^{T} \alpha_t = 1, \alpha_t > 0 \ \forall t$. The model merging

considers the problem where we have $T$ fine-tuned expert models $\{f_t(\boldsymbol{x}; \boldsymbol{\theta}_t)\}_{t=1}^T$ and pre-trained weight $\boldsymbol{\theta}_0$, composing a multitask model $\boldsymbol{\theta}^*$ to approximate the optimal solution.

$$\boldsymbol{\theta}_{opt} \approx \boldsymbol{\theta}^* = \mathcal{F}(\boldsymbol{\theta}_0, \boldsymbol{\theta}_1, \cdots, \boldsymbol{\theta}_T) \tag{1}$$

Here $\mathcal{F}$ represents an arbitrary merging function. For example, in Task Arithmetic [28], $\boldsymbol{\theta}^* = \boldsymbol{\theta}_0 + \sum_{t=1}^T \gamma_t(\boldsymbol{\theta}_t - \boldsymbol{\theta}_0)$.

Table 1: Merging without parameter interference and merging between similar tasks both cause performance degradation (Notice: these two experiments use different datasets).

| Task | Normalized Score (Equation (4)) |
|---|---|
| *With parameter interference* | |
| Fine-tuned | 100.00 |
| Merging | 85.43 |
| *Without parameter interference* | |
| Non-overlap Fine-tuned | 100.00 |
| Non-overlap Merging | 82.21 [↓ 3.21] |
| *Similar tasks* | |
| Fine-tuned | 100.00 |
| Similar-Tasks Merging | 91.58 [↓ 8.42] |

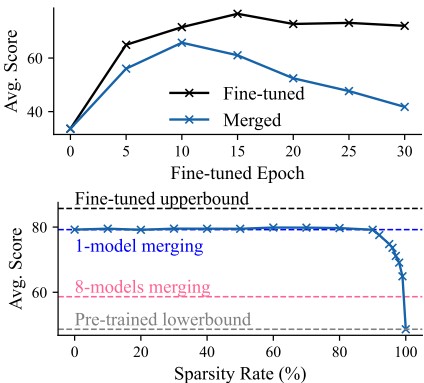

Figure 3: The impact of different ratios of shared knowledge and exclusive knowledge.

Although existing merging methods, like Task Arithmetic, can combine multiple task-specific models efficiently, they often exhibit significant performance gaps compared to single-task models. Prior research, such as Ties Merging [76], attributes this phenomenon to *parameter interference*. This term refers to the redundancy or sign discrepancies found in parameters located at the same position (e.g., self-attention weights) across different task models, which in turn result in information conflicting and performance loss. Additionally, *task interference*, as noted in multi-task learning literature [13, 31], arises from the inherent differences between tasks. For instance, tasks such as summarization, mathematical reasoning, and code generation require the model to process information in distinct ways. These differences worsen interference when models trained on different tasks are merged.

To understand these performance drops, we conducted two experiments using Task Arithmetic. First, we fine-tuned Qwen-14B with LoRA, assigning non-overlapping modules to avoid parameter interference. Despite this, a $3.21\%$ drop in performance occurred, indicating persistent interference. Second, using two similar summarization tasks (XSUM and DailyMail), we observed an $8.42\%$ drop compared to individually fine-tuned models, confirming that interference persists even between similar tasks. These results suggest that *interference in model merging is not limited to parameter-wise and task-wise issues.*

### 3.2 Interpreting Interference From the Perspective of Knowledge

To tackle the challenge of interference, we examine the merging process at a finer-grained knowledge perspective. We identify two types of critical knowledge: (1) *Shared knowledge*, which benefits multiple tasks, and (2) *Exclusive knowledge*, which is useful only for a specific task. Single-task models often contain both types, complicating the merging process and leading to interference. To validate our hypotheses, we conduct experiments that vary the ratio of task-specific and shared knowledge.

To examine the impact of shared knowledge, we conducted full fine-tuning on each model for its specific task. Excessive fine-tuning epochs can lead to catastrophic forgetting [19], a phenomenon where the model retains task-specific knowledge but loses general knowledge. As the fine-tuning epochs increase, the shared knowledge gradually decreases. The top section of Figure 3 illustrates that as the epoch count increases, merging performance significantly deteriorates, even though the fine-tuned model performs well on its task. This underscores the crucial role of shared knowledge in merging performance.

To explore the impact of exclusive knowledge, we merge a single task-specific model into the base model. We apply a sparsity method (*e.g.*, SVD) to reduce the ratios of task-specific weights in the

merging model from $100\%$ (standard merging) to $0\%$ (base model). As shown in the lower part of Figure 3, performance remains stable up to $90\%$ sparsity. Notably, even with a $99\%$ sparsity rate, a single-merged model outperforms multi-model merging, confirming the existence of exclusive knowledge, which is more pronounced with more models. This also underscores the value of unmerged task-specific knowledge, since the fine-tuning performance can be effectively restored by preserving unmerged task-specific information.

To summarize, both shared knowledge and un-merged task-specific knowledge play a vital role in merging performance. The exclusive nature of task-specific knowledge hinders the effectiveness of merging methods. Different types of knowledge need to be separated and modularized to achieve optimal performance. Thus, the first step of our Twin-Merging approach is to explicitly partition the weights into an expert containing shared knowledge and weights holding task-exclusive knowledge before merging. Formally, we denote the shared expert as $\boldsymbol{\theta}_s$ and the exclusive task-specific knowledge as $\{\boldsymbol{v}_t\}_{t=1}^T$, the detail of our method is illustrated in the following section.

### 3.3 Twin Merging

Our proposed Twin-Merging employs two main stages: **knowledge modularization** and **dynamic merging**. These stages are designed to narrow the performance gap and enhance adaptive knowledge composition. Building on the formulation in Equation (2), Twin-Merging preprocesses experts into shared experts, isolates and compresses exclusive knowledge into vectors, and dynamically composes them during inference.

The preprocess stage comprises three steps: (1) **Shared Expert**: To separate shared knowledge across different models, we consider the pre-merged model as a natural placeholder to encapsulate common knowledge that is important to all tasks (denoted as $\boldsymbol{\theta}^*$). By leveraging established merging techniques such as Task Arithmetic, we can readily extract the shared experts from the initial merged model. (2) **Exclusive Knowledge**: To convey task-specific information while separating common knowledge, we calculate the difference vector: $\boldsymbol{v}_t = \boldsymbol{\theta}_t - \boldsymbol{\theta}^*$. This subtraction vector preserves un-merged task-specific information while discarding the shared knowledge. (3) **Compressed exclusive vectors**: For practical use and distribution, we apply singular value decomposition (SVD) to further compress the above exclusive knowledge into vectors for each task. Assuming $\boldsymbol{v}_t$ has a rank-$m$ decomposition, $\boldsymbol{v}_t = \mathbf{U}_t \boldsymbol{\Sigma}_t \mathbf{V}_t^T$, we achieve a low-rank task space by selecting the top-$r$ singular values, resulting in $\mathbf{U}_t(r)\boldsymbol{\Sigma}_t(r)\mathbf{V}_t(r)^T$. We store only $\mathbf{U}_t(r), \boldsymbol{\Sigma}_t(r), \mathbf{V}_t(r)^T$.

---

**Algorithm 1** Twin-Merging

**Require:** language model $f(\boldsymbol{x}; \boldsymbol{\theta})$, pre-trained weight $\boldsymbol{\theta}_0$ and $T$ task-specific fine-tuned weights $\{\boldsymbol{\theta}_t\}_{t=1}^T$, trained router $\mathcal{R}$ parameterized by a full-connect layer $\boldsymbol{\phi}$, embedding $Emb$, compression rank $r$ and pre-specified weight $\{\gamma_t\}_{t=1}^T$

1: **Pre-calculation:**       ▷ Only excute once
2: Compute the shared expert $\boldsymbol{\theta}_s$:
3:    $\boldsymbol{\theta}_s \leftarrow \boldsymbol{\theta}_0 + \sum_{t=1}^T \gamma_t(\boldsymbol{\theta}_t - \boldsymbol{\theta}_0)$
4: Extract exclusive knowledge vectors for each task-specific weight:
5:    $\boldsymbol{v}_t \leftarrow \text{SVD}_r(\boldsymbol{\theta}_t - \boldsymbol{\theta}_s)$, for $t = 1, \ldots, T$

6: **Inference:**                ▷ Main loop
7: initialize output $\boldsymbol{Y}$
8: **for** each input $\boldsymbol{x}$ in inputs $\boldsymbol{X}$ **do**
9:    Calculate router weights:
10:     $[w_1, \cdots, w_T] \leftarrow \text{softmax}(\mathcal{R}(\text{Emb}(\boldsymbol{x}); \boldsymbol{\phi}))$
11:    Merge into a single expert $\boldsymbol{\theta}^*$:
12:     $\boldsymbol{\theta}^* \leftarrow \boldsymbol{\theta}_s + \sum_{t=1}^T w_t \boldsymbol{v}_t$
13:    Perform model inference to produce the output:
14:     $\boldsymbol{Y} \leftarrow \boldsymbol{Y} \cup f(\boldsymbol{x}; \boldsymbol{\theta}^*)$
15: **end for**

**Ensure:** Output $\boldsymbol{Y}$ for input $\boldsymbol{X}$.

---

In inference stage, adapting to unforeseen challenges is difficult, especially with varied test data. For example, if most of the data consists of a certain type (denoted as $\mathcal{D}_u$), we should tailor the merged model for that specific task to get the best results. Instead of pre-defining the best parameters, we propose a new approach that combines shared expertise with exclusive knowledge. Our method involves using the input $\boldsymbol{x}$ to dynamically adjust to the current data, enabling us to utilize shared knowledge and apply specialized expertise based on the inputs.

$$\boldsymbol{\theta}^* = \mathcal{F}(\ \underbrace{\boldsymbol{\theta}_s}_{\text{shared knowledge}},\ \underbrace{\boldsymbol{v}_1, \cdots, \boldsymbol{v}_T}_{\text{exclusive knowledge}},\ \boldsymbol{x}) \tag{2}$$

During inference, we fine-tune a small fuser $\mathcal{R}$ parameterized by $\boldsymbol{\phi}$ through empirical risk minimization on a small validation dataset. This fuser, trained to dynamically select the specific task experts,

replacing the need for complex optimization algorithms to determine fusion coefficients. The merging model is obtained by:

$$\boldsymbol{\theta}^* = \boldsymbol{\theta}_s + \sum_{t=1}^{T} w_t * \text{SVD}_r(\boldsymbol{\theta}_t - \boldsymbol{\theta}^*)$$

$$\{w_1, \cdots, w_T\} = \text{softmax}\left(\mathcal{R}(\text{Emb}(\boldsymbol{x}); \boldsymbol{\phi})\right)$$

(3)

Here, $\text{Emb}(\boldsymbol{x})$ represents the sequence of the last-layer token embeddings from the shared expert $f(\boldsymbol{x}; \boldsymbol{\theta}_s)$.

## 4 Experiments

### 4.1 Merging Experiment

**Baselines** We first compare Twin-Merging with several train-free model-merging methods on both discriminative and generative NLP benchmarks, including weight averaging, Task Arithmetic [28], Ties-Merging [76], and DARE Merging [79]. To compare with Merging methods that need validation dataset, we also conduct experiments on CV tasks with AdaMerging [78] and Surgery [77]. Details on these baselines are provided in Appendix D.

**Benchmarks** For language discriminative tasks, following [76, 79], we use RoBERTa [42] as the backbone and evaluate on the 8-task GLUE benchmark [69]. More details are in Appendix D.2. For language generative tasks, we use Qwen-14B [3] as the primary model to demonstrate the effectiveness of our approach on large-scale language models. To reduce deployment costs, we utilize task-specific checkpoints fine-tuned with the LoRA method [26] (See Appendix A for details on adapting Twin-Merging to LoRA). We evaluate our model on four scenarios: general knowledge (MMLU benchmark [24]), factualness (TruthfulQA [40]), safety (BBQ [52]), and summarization (CNN-DailyMail [48]).

For vision tasks, following AdaMerging [78], we use ViT-B/32 in CLIP [55] as the backbone on eight image classification datasets: SUN397 [75], Stanford Cars [35], RESISC45 [5], EuroSAT [23], SVHN [50], GTSRB [59], MNIST [10], and DTD [7]. We employ the best version of AdaMerging (layer-wise AdaMerging++) and the Surgery (AdaMerging version), and apply 90% sparsity for our Twin-Merging. Detailed information is provided in Appendix D.2.

**Metrics** We include individually fine-tuned models and the pre-trained model as upper and lower bounds on performance, respectively. To mitigate the effects of different task-specific score ranges, performance is assessed using the average normalized score of the fine-tuned models. The normalized score of merged model $\boldsymbol{\theta}^*$ is calculated as:

$$\text{Normalized Score} = \frac{1}{T} \sum_{t=1}^{T} \frac{\underset{x \sim \mathcal{D}_t}{\text{Score}}\left[f(\boldsymbol{x}; \boldsymbol{\theta}^*)\right]}{\underset{x \sim \mathcal{D}_t}{\text{Score}}\left[f_t(\boldsymbol{x}; \boldsymbol{\theta}_t)\right]}$$

(4)

Table 2: Performance on 8 Discriminative Tasks (RoBERTa) and 4 Generative Tasks (Qwen-14B)

| Method | 8 Discriminative Tasks | 4 Generative Tasks | Avg. |
|---|---|---|---|
| **Pretrained** | 41.69 | 91.06 | 66.37 |
| **Fine-tuned** | 100.00 | 100.00 | 100.00 |
| **Weight Averaging** | 52.56 | 95.74 | 74.15 |
| **Task Arithmetic** | 67.80 | 96.61 | 82.20 |
| **Task Arithmetic (w/ DARE)** | 64.66 | 98.52 | 81.59 |
| **Ties-Merging** | 63.68 | 92.67 | 78.17 |
| **Ties-Merging (w/ DARE)** | 65.58 | 91.92 | 78.75 |
| **Twin-Merging (Rank-1)** | 86.00 | 100.96 | 93.48 |
| **Twin-Merging (Ours)** | **96.14** | **102.38** | **99.26** |

Table 3: Performance and Cost on 8 CV Tasks (ViT-B/32)

| Method | Avg. Normalized Score | Additional Time Cost | VRAM |
|---|---|---|---|
| **Pretrained** | 52.02 | 18m48s | 3.6GB |
| **Fine-tuned** | 100.00 | 18m48s | 28.8GB |
| **Weight Averaging** | 72.30 | 18m50s | 3.6GB |
| **Task Arithmetic** | 76.50 | 21m34s | 3.6GB |
| **Ties-Merging** | 75.10 | 19m24s | 3.6GB |
| **AdaMerging** | 88.50 | 185m35s | 3.6GB |
| **Surgery** | 94.04 | 215m01s | 32.4GB |
| **Twin-Merging(Ours)** | **95.33** | 47m22s | 5.0GB |

**Main Results**  Table 2 presents the results for all discriminative and generative benchmarks for language tasks, while Table 3 provides the results for vision tasks. A comparison of each task is illustrated in Figure 2b, with detailed statistics provided in Table 8 and Table 9 in the Appendix D.7. We also list the full-finetuned LLaMA results in Appendix D.7.

For discriminative tasks, it approachs the upper bound of finetune performance in the GLUE benchmark. Specifically, our methods improve over Task Arithmetic by $28.34\%$, Ties-Merging by $32.46\%$, and DARE-Merging by $30.56\%$ in absolute normalized score. In Figure 2b, we observe that especially on the COLA task, where conventional merging methods fail to improve the result, our approach can still approach the upper bound of the COLA expert.

On generative tasks, Twin-Merging achieves strong performance, outperforming Task Arithmetic and DARE Merging by $5.77$ and $3.86\%$. Two interesting insights emerge: (1) The performance gains for Qwen-14B in generative tasks are smaller compared to RoBERTa in discriminative tasks. This indicates that smaller models like RoBERTa gain more from task-specific knowledge, while large models like Qwen-14B perform well because its strong general knowledge. (2)Twin-Merging surpasses the upper bound set by fine-tuned experts on the generative benchmark. This may be due to the extensive knowledge in Qwen-14B, where modularization and dynamic merging unlock further potential without additional fine-tuning. These findings highlight a promising path for improving large language models without retraining.

For vision tasks, Twin-Merging outperforms the AdaMerging and Surgery baselines with a higher accuracy ($95.33\%$ vs. $94.04\%$) while being more efficient in time and storage ($47m22s$ vs. $215m01s$, 5.0GB vs. 32.4GB). AdaMerging uses task-wise or layer-wise learnable parameters to improve merging, and Surgery adds task-specific modules after merging, requiring training on the validation set for all eight tasks. Surgery also needs prior knowledge of the task type before inference and involves multiple forward passes, leading to high VRAM usage. In contrast, our method efficiently handles diverse test inputs with minimal time and storage costs.

Table 4: Our method scalability (72B)

| Method | TruthfulQA | BBQ |
|---|---|---|
| **Pretrained-72B** | 94.48 | 89.51 |
| **Fine-tuned** | 100 | 100 |
| **Task Arithmetic** | 98.70 | 95.40 |
| **Twin Merging** | **99.30** | **97.14** |

Table 5: Performance (un-normalized[2]) on unseen tasks

| Method | QNLI+MNLI+RTE | MMLU |
|---|---|---|
| **Multi-Task Learning** | 44.63 | 63.74 |
| **Task Arithmetic** | 53.92 | 62.02 |
| **Task Arithmetic (w/ DARE)** | 54.27 | 63.09 |
| **Ties Merging** | 54.09 | 64.62 |
| **Ties Merging (w/ DARE)** | 54.72 | 63.13 |
| **Twin-Merging** | **55.86** | **65.98** |

**Scalability of Twin-Merging**  Our method remains effective with scaled models (*e.g.*, 72B parameters), as shown in Table 4. To manage high deployment costs, we limited our evaluation and merged experts to two tasks: BBQ and TruthfulQA. Twin-Merging consistently surpasses scaled pre-trained models and Task Arithmetic, highlighting our approach's scalability. Additionally, our method can be

---

[2]Notice that we cannot directly normalize them as we do not have the corresponding expert on unseen datasets to get upper-bound performances. This leads to relatively lower scores due to the narrower score ranges for tasks like RTE (max 66.43 vs max 91.71 for QNLI) and MMLU (max 68.03).

easily integrated with other merging methods, as detailed in Appendix D.9, making it both extensible and scalable.

## 4.2 Unseen Generalization

As shown in Table 5, Twin-Merging method benefits from complementary collaboration among different experts. Since the corresponding task-specific experts are unavailable, we directly use the average of the unnormalized scores as the metrics. In the GLUE benchmark, when QNLI, MNLI, and RTE experts are absent, our approach still outperforms traditional baselines. Details on the expert combination for QNLI can be found in Figure 5a. For complex tasks like MMLU, which involves multiple-choice QA tasks across 57 categories, Twin-Merging demonstrates superior performance using the combined knowledge from TruthfulQA, BBQ, and CNN-DailyMail domains.

## 4.3 Ablation Studies

Table 6: Ablation study of Twin-Merging

| Method | RoBERTa | Qwen |
|---|---|---|
| **Pretrain** | 41.69 | 91.06 |
| **Shared** | 67.80 | 96.61 |
| **Dynamic Merging** | 81.47 | 87.77 |
| **Pretrain + Dynamic Merging** | 85.90 | 95.03 |
| **Shared + Dynamic Merging (Twin Merging)** | 96.14 | 102.38 |

To demonstrate the effectiveness of our approach, we conducted ablation studies for Twin-Merging, summarized in Table 6. Removing dynamic experts from the Shared model leads to a significant performance loss (96.14 vs. 67.80), highlighting the need for dynamic merging. Replacing the shared expert with a task-specific expert also results in a clear drop in performance (96.14 vs. 81.47), showing the value of the shared expert in capturing common knowledge.

Additionally, applying dynamic merging directly to a pretrained model performs worse than Twin Merging (85.90 vs. 96.14), likely due to two factors: (1) Pretrained models may lack rich task knowledge, while the shared expert in Twin Merging captures diverse, task-specific knowledge. (2) Subtracting the pretrained model fails to fully consider exclusive knowledge specific to each task, leading to interference, as analyzed in Section 3.2.

**Discussion 1** We find that removing dynamic experts severely impacts RoBERTa but has less effect on Qwen-14B, suggesting that smaller models rely more on task-specific biases, while larger models benefit more from general shared knowledge. This indicates that our method adapts effectively to the varying knowledge requirements of models of different sizes.

**Discussion 2** Compared to simpler routing methods like *direct route to task-specific expert* or *combining multiple experts* based on multiple LoRA [27, 81], Twin Merging delivers better performance, especially on unseen tasks, by reducing interference and leveraging complementary knowledge.

*Direct route to task-specific expert* refers to the fine-tuned baseline in Table 2. This approach assumes perfect routing and the absence of out-of-domain data, where each task uses its own dedicated expert. It represents the ideal scenario and serves as an oracle baseline to highlight the performance gap for merging methods. Despite this, Twin Merging still improves performance on generative tasks (102.38 vs. 100.0) and unseen tasks (Table 5) by leveraging different sources of exclusive knowledge. Moreover, this baseline demands storing all task-specific experts, which significantly increases storage, as discussed in Section 4.6.

In *combining multiple experts*, the lack of separation between shared and exclusive knowledge leads to interference, as conflicts between exclusive knowledge are inevitable (Section 3.2). There are two ways to combine experts: (1) Static Combination: This is akin to "Task Arithmetic" in LoRA (Table 2). Twin Merging outperforms static combinations (102.38 vs. 96.61). (2) Dynamic Combination: This matches "Pretrain + Dynamic Merging" method in Table 6, and Twin Merging again shows superior performance (102.38 vs. 97.03).

## 4.4 Scale to More Tasks

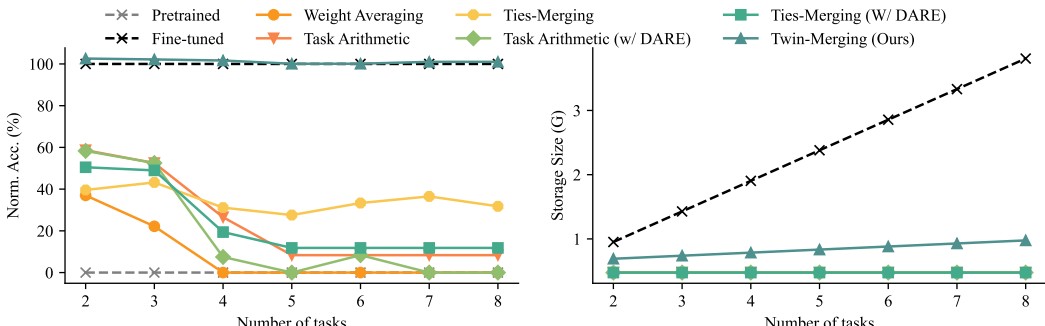

Figure 4: Averaged normalized accuracy *vs.* the number of tasks for various benchmarks. Twin-Merging maintains performance regardless of task number and compresses the fine-tuned checkpoints.

In the left panel of Figure 4, we examine the impact of the number of tasks on model merging performance. Conventional model merging methods degrade notably, especially with many tasks, nearly reaching pre-trained levels. However, Twin-Merging consistently outperforms other methods, approaching fine-tuned performance, with greater gains as the task count rises.

The right panel of Figure 4 shows the performance-storage trade-offs. While model merging methods have a constant storage cost, their performance remains low. In contrast, maintaining individual task-specific models guarantees strong performance but requires excessive storage. Twin-Merging achieves nearly 100% normalized accuracy across various tasks, balancing performance and storage efficiency by maintaining task-specific parameters with shared experts. This makes Twin-Merging a viable solution for scenarios demanding a balance between performance and storage efficiency.

## 4.5 Router Analysis

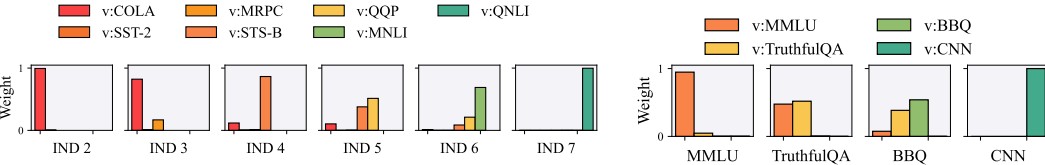

(a) The routing result on the QNLI dataset using different numbers of GLUE experts, ranging from 2 twin ($v_{\text{CoLA}}$ and $v_{\text{SST-2}}$) to 7 twin vectors ($v_{\text{CoLA}}$, $v_{\text{SST-2}}$, $v_{\text{MRPC}}$, $v_{\text{STS-B}}$, $v_{\text{QQP}}$, $v_{\text{MNLI}}$, and $v_{\text{QNLI}}$). The router weights are Softmax normalized.

(b) The routing weight of Qwen experts ($v_{\text{MMLU}}$, $v_{\text{TruthfulQA}}$, $v_{\text{BBQ}}$, $v_{\text{CNN-DailyMail}}$) on four generative tasks (MMLU, TruthfulQA, BBQ, CNN-DailyMail).

Figure 5: Twin-Merging routing decisions of the experts for various tasks.

Figure 5 shows the results of routing decisions among experts for the QNLI dataset and four generative benchmarks. As shown in Figure 5a, the router maximizes the use of limited expert knowledge to address QNLI, a task where the goal is to determine if the context sentence contains the answer to the input question. For example, with only $v_{\text{CoLA}}$ and $v_{\text{SST-2}}$ available, the router primarily uses $v_{\text{CoLA}}$, which provides knowledge of sentence and word relations, while $v_{\text{SST-2}}$ is focused on irrelevant sentiment classification. With six experts ranging from $v_{\text{CoLA}}$ to $v_{\text{MNLI}}$, the router mainly leverages $v_{\text{MNLI}}$ for textual entailment and $v_{\text{QQP}}$ for question-answering capabilities. When $v_{\text{QNLI}}$ is included, the router naturally relies on QNLI-specific knowledge. These results demonstrate the flexibility and adaptability of our Twin-Merging method, providing good interpretability. For larger models like Qwen-14B, as shown in Figure 5b, the router plays a crucial role in selecting and combining specific knowledge. When experts have overlapping task-specific knowledge, such as $v_{\text{TruthfulQA}}$ and $v_{\text{MMLU}}$, the router may assign them similar weights.

## 4.6 Compression and Speed Analysis

**Compression Analysis** In the left panel of Figure 6, we explore sparsity rates from 0% to 100%. Appendix E attachs detail qualtivie analysis of various Merging methods. Remarkably, our Twin-

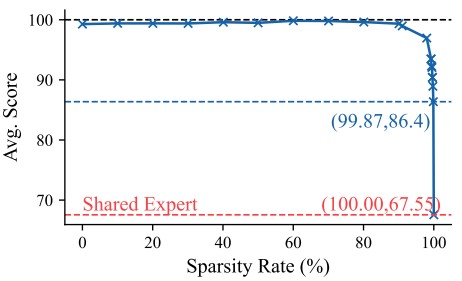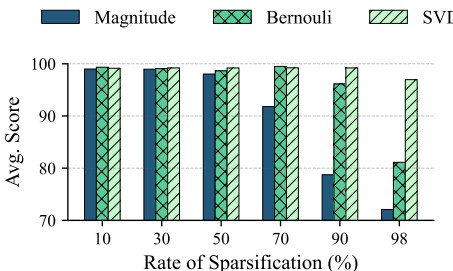

Figure 6: Twin-Merging performance *vs.* different sparsity levels and techniques for GLUE

Merging method maintains $86.4\%$ performance even at a $99.8\%$ compression rate. This suggests that performance relies on a small fraction of task-specific parameters, aligning with previous findings [76, 79]. Our results also validate our hypothesis that redundant parameters can obscure critical knowledge, leading to performance degradation. Consequently, we primarily use a $90\%$ sparsity rate in our experiments to preserve performance while reducing storage costs. We also conducted an ablation study on sparsity methods, shown on the right side of Figure 6. SVD better retains task-specific information compared to Magnitude [76] and Bernoulli Dropout [79]. As SVD is applied only once during preprocessing, it does not become an inference bottleneck.

Table 7: Compute-performance tradeoff in the generative benchmark.

| Method | Training Tokens | Training Cost | Inference Cost (/1000 items) | Performance |
|---|---|---|---|---|
| **Multi-Task Learning** | 536.35M | 10h32min | 236s | 94.31 |
| **Model Merging** [3] | 0 | 0 | 236s | 96.61 |
| **Twin-Merging** | 0.57M | 183s | 275s | 102.38 |

**Speed Analysis** Table 7 presents the time cost for Twin-Merging in generative benchmarks. Although the training stage utilizes only 0.1% of the total training budget, Twin-Merging significantly improves general capabilities compared to multi-task learning. Compared to conventional model merging methods, Twin-Merging sacrifices minimal router training budget and incurs a slight reduction in inference speed for dynamically composing the twin vectors, thereby achieving superior performance. More detailed analysis and results are provided in Appendix E. In summary, our approach strikes a better balance between computational cost and performance.

# 5   Conclusions

In this paper, we introduce the Twin-Merging to merge language models, aiming to close the performance gap between conventional model merging techniques and fine-tuned models, while improving adaptability to data heterogeneity. By modularizing and dynamically merging shared and task-specific knowledge, Twin-Merging significantly outperforms existing model-merging methods and approaches the performance of fine-tuned models across various settings and domains. Our study highlights the impact of shared and exclusive task-specific knowledge on merging performance. We show that Twin-Merging benefits even strong scaled models like Qwen-72B, which already perform well across domains. It extends to more tasks and merging methods, demonstrating better generalization on unseen data. By utilizing SVD, our solution retains $86\%$ of the performance with only $0.1\%$ of the parameters, approaching upper-bound performance with minimal storage increase as tasks grow, achieving a better tradeoff between computation and performance.

# 6   Acknowledgments

We thank the Shanghai AI Laboratory for supporting GPU resources. We also thank the anonymous reviewers for their comments on improving the quality of this paper and Netmind.AI for their resource/technical support.

---

[3]Here, we assume that merging method does not retrain all task experts; instead, it reuses experts (*e.g.*, downloaded from model hubs like Huggingface [73]).

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

## A  Twin Merge on LoRA

Here, we will demonstrate that our Twin-Merging method can be seamlessly applied to LoRA module [26], where the base model is fixed and additional task-specific information is injected through matrix, *i.e.*, $\boldsymbol{\theta}_t = \boldsymbol{\theta}_0 + \text{LoRA}_t$, where $\text{LoRA}_t$ represents the fine-tuned LoRA module for the $t$-th task. let $\boldsymbol{\theta}_s = \boldsymbol{\theta}_0 + \text{LoRA}_s$, we can prove that Twin-Merging on the $\boldsymbol{\theta}$ is equivalent to Twin-Merging on the LoRA module.

$$
\begin{aligned}
\boldsymbol{\theta}^* &= \underbrace{\boldsymbol{\theta}_s + \sum_{t=1}^{T} w_t * \text{SVD}_r(\boldsymbol{\theta}_t - \boldsymbol{\theta}_s)}_{\text{Twin-Merging on } \boldsymbol{\theta}} \\
&= \boldsymbol{\theta}_0 + \text{LoRA}_s + \sum_{t=1}^{T} w_t * \text{SVD}_r\bigg( (\boldsymbol{\theta}_0 + \text{LoRA}_t) - (\boldsymbol{\theta}_0 + \text{LoRA}_s) \bigg) \\
&= \boldsymbol{\theta}_0 + \underbrace{\text{LoRA}_s + \sum_{t=1}^{T} w_t * \text{SVD}_r(\text{LoRA}_t - \text{LoRA}_s)}_{\text{Twin-Merging on LoRA}} \\
&= \boldsymbol{\theta}_0 + \text{LORA}^*
\end{aligned}
\tag{5}
$$

where we denote $\text{LORA}^* = \text{LORA}_s + \sum_{t=1}^{T} w_t * \text{SVD}_r(\text{LoRA}_t - \text{LoRA}_s)$.

## B  More relative research

**Multi-Task Learning.**   The multi-task training typically learns multi-task features by simultaneously optimizing task-specific objectives, facilitating the integration of diverse knowledge into the model. Existing works mainly focus on mitigating task conflicts [41] and catastrophic forgetting [19] by parameter sharing [45], adjusting suitable objectives [14, 57], find suitable task weighting [4, 49], and minimizing negative transfer [31]. In an era where models are growing larger, and the number of task scenarios is increasing, what we need to explore is a more cost-effective approach to multi-task learning. Therefore our focus is on multi-task scenarios that do not require acquiring or integrating multi-task data and do not involve additional updates to existing experts.

**Mixture of Experts.**   To enhance model scalability without increasing computational costs, the mixture of experts (MoE) paradigm introduces conditional routing of inputs to a subset of learnable parameters. Several efforts have extended feedforward networks (FFNs) within Transformers to incorporate MoE layers, such as GShard [36] and Switch Transformer [17]. These models typically employ learnable top-2 or top-1 routing strategies to scale MoE language models to an extremely large size [30]. Recent studies have focused on challenges such as load balancing of experts [8, 82], training instability [86], expert specialization [9, 66], and synchronization reduction [64]. However, these methods often require substantial multi-task data and costly joint training. In contrast, our approach directly reuses task-specific experts, leading to the natural specialization of experts in different domains. We only require minimal fine-tuning for a small router to calculate fusion weights, making our method highly efficient.

## C  The Merging Interference and Limited Generalization

To illustrate the challenge in determining the optimal merging coefficient and the limitations of pre-specified coefficients with unpredictable data, we consider COLA and SST-2 as in-domain experts. We merge them using Task Arithmetic and evaluate on the eight discriminative tasks from the GLUE benchmark. Only COLA and SST-2 are seen tasks, while the others are unseen. Since the merging coefficient is crucial for performance [51, 78], we conduct an extensive grid search for coefficients ranging from $-2$ to $2$.

A large dark-blue region indicates consistent optimal performance, which is why Task Arithmetic can work with various weights. Conventional methods search this region for optimal performance across

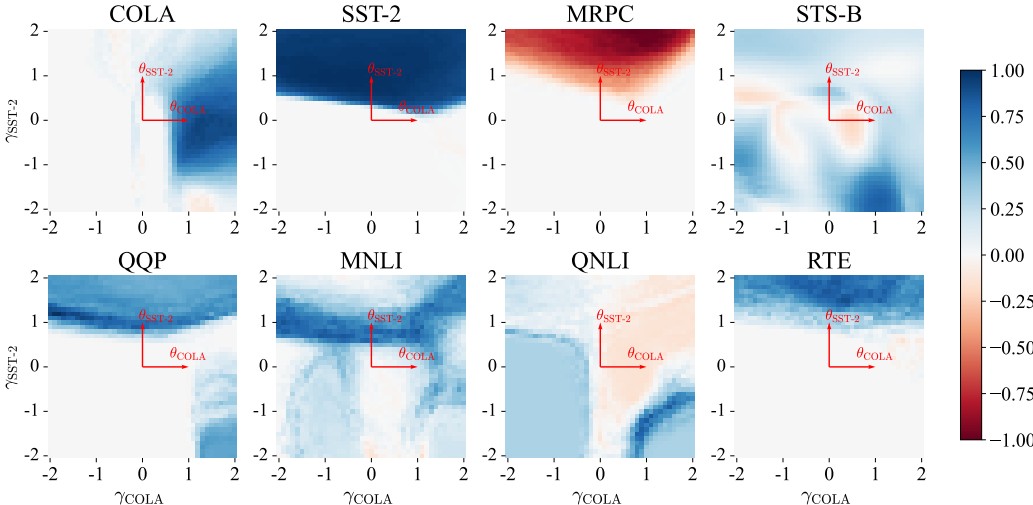

Figure 7: The visualizations show normalized performance across eight GLUE tasks, highlighting the impact of combining expertise from the COLA and SST-2 domains (expert indicated by red vectors) through Task Arithmetic. Performance scores are normalized, with the unmerged pretrained model set to zero and other results scaled to the $[-1, 1]$ range. The x-axis ($\gamma_{\text{COLA}}$) and y-axis ($\gamma_{\text{SST-2}}$) represent the merging weights for COLA and SST-2 expertise. Blue regions indicate improved performance over the pretrained model, while red regions indicate deterioration.

all in-domain tasks, avoiding the red region. However, this is computationally expensive and does not scale well with an increasing number of tasks. Additionally, it cannot handle unseen tasks, as the same coefficients can produce different patterns across tasks. For example, setting coefficients $\gamma_{\text{COLA}}$ and $\gamma_{\text{SST-2}}$ to 1 leads to performance drops in MRPC and QNLI, but gains in MNLI, QQP, and RTE. [4]

Furthermore, merging performance is not always a single cluster. For example, within the range of $[-2, 2]$, STS-B and QNLI already show complex patterns, making it difficult to find an optimal weight for all tasks when task-specific experts are limited. Although Yang et al. [78] propose unsupervised entropy minimization to find optimal coefficients, this method is limited to classification tasks and has limited adaptability.

To address this, we propose reformulating the problem of fusing models as a supervised learning task. Specifically, we train a router to dynamically merge task-specific experts, as detailed in Section 3.3.

# D  Experiment Details

Here we detaily illustrate the setting of our experiments.

## D.1  Compute Resources Used and Runtimes

We executed all our experiments on Nvidia A100 GPUs equipped with 80GB RAM. Single-task LoRA models for Qwen-14B on four generative tasks required 1-2 hours per task, Single-task LoRA for Qwen-72B need 10 hours on single GPUs to train. while the multitask vector took around 10 hours on single GPUs of 500M tokens. The RoBERTa model needs 15 minutes per task on GLUE datasets.

Our router is implemented as a three-layer linear network with Leaky ReLU activations and batch normalization. We train the router on the validation dataset with a learning rate of 5e-4 for 10 epochs. The validation set consists of the integration of in-domain downstream tasks, not the general text corpus. The validation set is taken from a split of the training set, and we use at most 1,000 items for router training for each task.

---

[4] In fact, the MNLI and QNLI are very similar tasks about Natural Language Inference (NLI) [69]. This demonstrates that task similarity does not guarantee similar merging performance patterns.

Merge experiments were efficient, with evaluations consuming less than 2 minutes. The inference is generally fast within 4 minutes per 1000 items for generative tasks and less than 30 seconds per 1000 items for discriminative tasks. The detail comparison of the training cost and inference cost of different methods are detailed in Table 7.

### D.2 Employed Datasets and Associated Licences

**Discriminative Tasks.** we conduct experiments on the GLUE benchmark [69] with eight discriminative tasks, which is designed for classification tasks except for STS-B for the regression task. The detail of eight dataset can be found in the paper of Wang et al. [69]. Consistent with prior research [79], We split 10% of the training set as a validation set and employ the original validation data as the test set.

The licenses of QNLI, COLA, and STS-B are licensed under CC-BY-SA. QQP is licensed under MIT. SST-2 and MRPC are licensed under Apache 2.0. MNLI is licensed under OANC. RTE is licensed under CC BY 4.0. Thus, these datasets in GLUE are available for non-commercial research purposes.

**Generative Tasks.** We conducted experiments on four benchmarks:

1. **MMLU** [24]: This benchmark tests general and STEM knowledge across 57 subjects, from elementary to professional levels. We used Exact-Match as the metric.

2. **TruthfulQA** [40]: This benchmark assesses the truthfulness of language models with 817 questions spanning 38 categories like health, law, finance, and politics. Exact-Match was used as the metric.

3. **BBQ** [52]: This dataset highlights social biases against protected classes in nine social dimensions relevant to U.S. English-speaking contexts. Exact-Match was the metric.

4. **CNN-DailyMail** [48]: This dataset is used for text summarization, requiring models to generate summaries of news stories. ROUGE-2 scores [39] were used for evaluation.

We evaluated these tasks using the HELM benchmark[5] in a few-shot setting.

For MMLU and TruthfulQA, which lack official training sets, we used the Dolly-15k dataset[6] for MMLU and the BigBench-sampled dataset for TruthfulQA.

The GSM8K and MMLU datasets are under the MIT License. TruthfulQA and CNN-DailyMail are under the Apache-2.0 License. BBQ is under the CC-BY 4.0 License. These datasets are available for non-commercial research purposes.

**Vision Tasks.**

1. **SUN397** [75]:A scene classification dataset containing 108,754 images across 397 classes, with each class having at least 100 images.

2. **Stanford Cars** [35]: A car classification dataset comprising 196 classes and a total of 16,185 images, divided equally between the training and test sets.

3. **RESISC45** [5]: A remote sensing image scene classification dataset with 45 classes and 31,500 images, approximately 700 per class.

4. **EuroSAT** [23]: A satellite image classification dataset consisting of 27,000 labeled and geo-referenced images across 10 classes.

5. **SVHN** [50]: A real-world digit classification dataset extracted from Google Street View images. It includes 10 classes, with 73,257 training samples, 26,032 test samples, and 531,131 additional simple samples.

6. **GTSRB** [59]: A traffic sign classification dataset containing over 50,000 images across 43 classes of traffic signs.

7. **MNIST** [10]: A benchmark dataset for image classification, containing grayscale images of handwritten digits across 10 classes. The training set has 60,000 images, and the test set has 10,000 images, with a balanced distribution among classes.

---

[5]https://github.com/stanford-crfm/helm
[6]https://huggingface.co/datasets/databricks/databricks-dolly-15k

8. **DTD** [7]: A texture classification dataset with 47 classes and a total of 5,640 images, with approximately 120 images per class.

### D.3 Language Model Backbone

For discriminative tasks, we used RoBERTa-base[7] [42] as our pre-trained backbone and fine-tuned it for each dataset to create supervised models. We conducted separate fine-tuning for the RoBERTa-base model on each dataset for 10 epochs. Our selected hyperparameters included a batch size of 64 and a learning rate set at $1e^{-5}$.

For generative tasks, we employed Qwen-14B[8] as the backbone and applied LoRA [26] for task-specific fine-tuning. In the case of generative tasks, the fine-tuning process for Qwen-14B involved the utilization of LoRA with a rank set to 32, a batch size of 128, and a learning rate of $2e^{-4}$ for 3 epochs. For Qwen-72B we employ the same setting with QLoRA technique [11].

### D.4 Non-Overlapping Merging

To serperate the impact of parameter-wise interference, we design the non-overlapping experiment based on Qwen LoRA modules as follows: (1) Firstly, we obtain standard merging experts by injecting the LoRA module into both the "w1" and "c_proj" weights of the Qwen-based model, and fine-tune them on two different tasks, resulting in two distinct models. Then we combine it into a single model to obtrain standard merging results. (2) Next, we perform a non-overlapping fine-tuning by injecting LoRA only to "w1" on one task and "c_proj" on another, producing two models with task-specific knowledge in different modules. (3) Finally, we combined the non-overlapping checkpoints to get the merged results. Since task-specific knowledge was injected into separate modules, parameter-wise interference was minimized. The results are shown in the upper section of Table 1.

### D.5 Sparsification Methods Details

In Figure 6, we conduct a comparative analysis employing various sparsification methods. The specifics of each method are outlined below:

- **Magnitude**. Following the setting in Ties-Merging [76], we retain solely the $k\%$ largest-magnitude values while resetting the remaining values to zero.

- **Bernoulli-Dropout**. Adhering to the methodology introduced in DARE [79], we employ a parameterized Bernoulli distribution to sample a sparse mask $\boldsymbol{m}^t$. This mask is then applied to the parameters $\boldsymbol{\delta}$ and subsequently rescaled with respect to the mask rate $k$.

$$
\begin{aligned}
\boldsymbol{m}^t &\sim \text{Bernoulli}(k), \\
\widetilde{\boldsymbol{\delta}}^t &= \boldsymbol{m}^t \odot \boldsymbol{\delta}^t, \\
\hat{\boldsymbol{\delta}}^t &= \widetilde{\boldsymbol{\delta}}^t/(1-k).
\end{aligned}
\tag{6}
$$

- **Singular value decomposition (SVD)**. Assuming that matrix $M$ has a rank-$m$ decomposition, expressed as $\mathbf{M} = \mathbf{U}_t\boldsymbol{\Sigma}_t\mathbf{V}_t^T$ where $\mathbf{U}_t \in \mathbb{R}^{d_{out} \times m}, \boldsymbol{\Sigma}_t \in \mathbb{R}^{m \times m}, \mathbf{V}_t \in \mathbb{R}^{d_{in} \times m}$. We compress the matrix $\mathbf{M}$ by selecting only the top-$r$ singular values from $\boldsymbol{\Sigma}_t$, denoted as $\mathbf{M}_r = \mathbf{U}_t(r)\boldsymbol{\Sigma}_t(r)\mathbf{V}_t(r)^T$. Here, $\mathbf{U}_t(r) \in \mathbb{R}^{d_{out} \times r}, \boldsymbol{\Sigma}_t(r) \in \mathbb{R}^{r \times r}, \mathbf{V}_t^r \in \mathbb{R}^{d_{in} \times r}$ represent sub-matrices of $\mathbf{U}_t, \boldsymbol{\Sigma}_t, \mathbf{V}_t^T$. This transformation significantly reduces the task-specific parameter dimensionality from $m \times (d_{out} + d_{in} + 1)$ to $r \times (d_{out} + d_{in} + 1)$, as the maximum $m$ typically equals to the hidden size of the language model (*e.g.*, $m = 768$ for RoBERTa-base and $m = 4096$ for Qwen-14B) and $r$ can be reduced to 1, resulting in a significant reduction in parameters and storage effectiveness.

  During merging, we decompress these matrices by extending $\mathbf{U}_t(r), \boldsymbol{\Sigma}_t(r), \mathbf{V}_t(r)^T$ to size-$m$ by filling with zeros, allowing us to recover $\mathbf{M}$ via their product. This operation is only at the matrix level; once we obtain the merged matrix, we discard the decompressed matrices, ensuring efficient storage.

---

[7] https://huggingface.co/FacebookAI/roberta-base
[8] https://huggingface.co/Qwen/Qwen-14B

## D.6 Baselines Details

Here we will elaborate on the baselines utilized in our main comparison experiment, as outlined in Table 2 and Figure 2b.

- **Individual** means that each task uses the corresponding fine-tuned model, which has no interference between tasks but cannot perform multiple tasks simultaneously. It serves as the upper-bound performance for each specific task.
- **Weight Averaging** [6, 74] is the simplest form of model merging, which straightforwardly averages the parameters of multiple models. It serves as a lower bound for model merging.
- **Task Arithmetic** [28] first introduces the concept of "task vectors" and merges them into the pre-trained model to execute multi-task learning.
- **Ties-Merging** [76] addresses task conflicts by eliminating redundant parameters. The process involves three steps: Trim, Elect Sign, and Disjoint Merge.
- **Task Arithmetic (w/ DARE)** [79] This variant incorporates the Bernoulli-Dropout technique for 70% sparsification before employing Task Arithmetic [28] for merging.
- **Ties-Merging (w/ DARE)** [79] Similar to the previous approach, this variant integrates Bernoulli-Dropout for 70% sparsification, followed by Ties-Merging [76] for the merging process.
- **AdaMerging** [78] assumes access to an offline test set and dynamically adapts to it by introducing additional coefficients at every layer, conducting unsupervised training across multiple iterations on the test set (without labels) to refine the model.
- **Surgery** [77] assumes that test data IDs are accessible during inference, allowing it to insert corresponding task-specific adapters to leverage task-specific knowledge.

The coefficient for Task Arithmetic and Ties-Merging are decided by a small scale grid search on validation datasets. The coefficient of 0.7 is consistently applied for DARE Merging, following the previous papers [79].

## D.7 Detail Results

In Table 2, we present only the average normalized scores across various tasks. In this section, we detail the statistical performance of all tasks, with discriminative results displayed in Table 8 and generative results shown in Table 9.

Table 8: The detail statistics of different merging performance on 8 discriminative tasks. **Bold** numbers indicate the best-averaging performance across different model merging methods.

| Model | COLA | STS-2 | MRPC | STS-B | QQP | QNLI | MNLI | RTE | Avg. |
|---|---|---|---|---|---|---|---|---|---|
| Pre-trained | 0.00 | 53.76 | 85.01 | 4.01 | 37.48 | 53.05 | 37.09 | 71.19 | 41.69 |
| Fine-tuned | 100.00 | 100.00 | 100.00 | 100.00 | 100.00 | 100.00 | 100.00 | 100.00 | 100.00 |
| Weight Averaging | 0.00 | 59.21 | 85.79 | 46.99 | 45.37 | 63.94 | 48.00 | 71.19 | 52.56 |
| Task Arithmetic | 8.35 | 88.26 | 89.57 | 32.84 | 82.03 | 85.40 | 75.54 | 80.43 | 67.80 |
| Ties-Merging | 31.76 | 88.86 | 86.18 | 10.94 | 61.05 | 85.94 | 83.01 | 69.56 | 64.66 |
| Task Arithmetic (w/ DARE) | 0.00 | 88.14 | 86.61 | 30.19 | 84.33 | 79.09 | 63.95 | 77.16 | 63.68 |
| Ties-Merging (w/ DARE) | 11.82 | 95.52 | 85.75 | 9.43 | 86.77 | 88.67 | 83.13 | 63.59 | 65.58 |
| Twin-Merging (Rank-1) | 51.24 | 98.67 | 89.20 | 76.31 | 92.16 | 93.24 | 96.45 | 90.76 | 86.00 |
| Twin-Merging (90% compressed) | **101.01** | **99.88** | **99.41** | **79.89** | **99.14** | **99.67** | **96.68** | **93.47** | **96.14** |

We primarily merge using LoRA for the Qwen-14B models because fully finetuning them to obtain task-specific experts would require a huge amount of resources and computation (finetuning the full 14B model requires at least 8 A100 GPUs). However, to further demonstrate, we provide experiments on fully fine-tuned LLaMA 7B models for generative tasks (GSM8k and TruthfulQA), where our approach still exhibits superior performance.

## D.8 More Advantages of Our Method

Beyond its strong performance and efficiency, our method offers key benefits in handling distribution shifts and LLM deployment, surpassing approaches like AdaMerging and Surgery.

Table 9: The detail statistics of different merging performance on 4 generative tasks. **Bold** numbers indicate the best-averaging performance across different model merging methods. Underlines indicate the second best performance of each task across different model merging methods.

| Model | MMLU | TruthfulQA | BBQ | CNN-DailyMail | Avg. |
|---|---|---|---|---|---|
| **Pretrained** | 101.37 | 94.35 | 86.27 | 82.24 | 91.06 |
| **Fine-tuned** | 100.00 | 100.00 | 100.00 | 100.00 | 100.00 |
| **Weight Averging** | 99.63 | 92.04 | 88.01 | 103.28 | 95.74 |
| **Task Arithmetic** | 98.93 | 98.23 | 83.65 | 105.62 | 96.61 |
| **Task Arithmetic (w/ DARE)** | 99.22 | 96.90 | 88.56 | 109.40 | 98.52 |
| **Ties-Merging** | 99.88 | 92.04 | 89.92 | 88.83 | 92.67 |
| **Ties-Merging (w/ DARE)** | **101.41** | 97.66 | 86.81 | 81.80 | 91.92 |
| **Twin-Merging (rank-1)** | 99.40 | 95.58 | 93.46 | **115.39** | 100.96 |
| **Twin-Merging (rank-16)** | 99.87 | **98.23** | **97.00** | 114.43 | **102.38** |

Table 10: Performance of LLaMA-7B

| Method | Avg. | Inference Time (/1000 items) |
|---|---|---|
| **Task-Arithmetic** | 69.89 | 186s |
| **Twin Merging** | 88.18 | 198s |

AdaMerging tackles distribution shifts in image domains by requiring access to test sets, even in unseen domains. This is problematic for online settings where test data is unpredictable. In offline scenarios, scaling test sets to large sizes is inefficient. Additionally, AdaMerging's entropy-based optimization limits it to classification tasks, which restricts its applicability as generative models like LLaMA and GPT-4 become more prominent.

Our method overcomes these limitations by dynamically adapting to test inputs, as shown in Table 5. It does not require specific validation datasets for each in-domain expert, making it more flexible. For example, the open-source Dolly dataset can represent the MMLU expert, and we do not need validation sets for QNLI, MNLI, RTE, or MMLU. Instead, we assume that combining in-domain knowledge helps address out-of-domain inputs, as supported by [71]. This dynamic merging ensures better generalization to diverse inputs, unlike AdaMerging, which relies on entropy approximations without supervision.

Our approach is also optimized for real-world LLM deployment with continuous data streams and no gradient updates, aligning with current trends [54]. Key advantages include:

- **Handling Unpredictable, Heterogeneous Data**: Our dynamic merging technique efficiently addresses diverse streaming inputs.

- **Reducing Latency and Storage**: We shift router training and knowledge modularization to preprocessing, and apply SVD techniques to minimize storage needs.

- **Broad Applicability**: Our method works across NLP and CV tasks, including generative tasks with Qwen-14B and up to 72B models, supporting large-scale AI deployment.

## D.9 Compatibility of Twin-Merging with Other Merging Methods

Table 11: Our method extensibility to other model merging methods

| Method | RoBERTa | Qwen |
|---|---|---|
| **Weight Average** | 52.56 | 95.74 |
| **Twin-Merging + Weight Average** | **96.23** | **100.08** |
| **Task-Arithmetic** | 67.80 | 98.52 |
| **Twin-Merging + Task-Arithmetic** | **96.14** | **102.38** |
| **Ties-Merging** | 63.68 | 92.67 |
| **Twin-Merging + Ties-Merging** | **96.34** | **102.35** |

To evaluate the compatibility of Twin-Merging with other merging methods, we conducted experiments using different techniques to create a shared expert, followed by dynamically merging the twin vectors. The results in Table 11 demonstrate that our method integrates seamlessly with primary merging techniques, leading to significant improvements. For example, when combined with our approach, the baseline Weight Average method improves from 52.26 to 96.23 on GLUE, approaching the performance of fine-tuned experts. Notably, our method complements Ties-Merging particularly well, suggesting that better isolation of shared knowledge enhances the overall performance of Twin-Merging.

# E   Inference Efficiency

Assume we have $T$ tasks, the fine-tuned model have $P = P_f + P_a$ parameters, where $P_f$ are frozen and $P_a$ are activated.

**Parameter Count and Storage Cost**   Assuming each float parameter uses 16 bits (either fp16 or bf16): Fine-tuned models require $2(TP_a + P_f)$ bytes of storage. Pretrained models, including those using Weight Average, Task Arithmetic, Ties-Merging, and DARE Merging techniques, each need $2P$ bytes of storage per model. For Twin-Merging, with the router having $P_r$ parameters ($P_r \ll P$) and a compression rate of $k\%$, it need to store $2TkP_a + 2P + P_r$ bytes including a shared expert, compressed exclusive task-specific vectors, and the router. We can select $k$ to compress the model matrix to rank $1$ for best storage. These strategies enhance the accessibility and sustainability of task-specific models, fostering wider advancements and applications. Visual representations can be found in Figure 2a and Figure 4.

**Computation FLOPs Analysis**   Our method mainly introduce the extra time cost due to routing and dynamical merging. However, as the inference process typically involves hundreds of forward passes (*e.g.*, 300 tokens for summarization tasks), the additional computing is usually neglectable. Assuming context length $s$, task number $T$, layer number $m$, the introduced FLOPs ( Multiply–accumulate operation ) can be computed as $m(24sh^2 + 4bs^2h)$ for routing, $Tm(12h^2 + 9h)$ for merging (excluding norm parameter), while generating $L$ tokens typically requires $\sum_{l=s}^{L} 24m(lh^2 + 4bl^2h)$ FLOPs. Given that $n \ll L$ and $s$ are typically truncated, the additional consumption is neglectable in generation tasks. We demonstrate the actual time cost in Table 7, which adds only 0.039 seconds per sample while bringing significant performance improvements.

**Comparison with other baselises**   Moreover, our approach offers significant performance improvements with these additional computing resources. As shown in Table 2 and the "More Baseline" section, we achieve an absolute normalized improvement of 28.34% for RoBERTa, 18.83% on ViT-B-32 compared to Task Arithmetic, 9.71% compared to Twin-Merging on Qwen-14B. Traditional model merging methods often overlook the heterogeneous nature of test inputs, leading to substantial performance gaps. Advanced merging techniques like AdaMerging and Surgery typically require costly training and searching processes, as demonstrated in Table 3. In contrast, our method achieves superior performance to fine-tuned models with minimal cost and storage requirements ($47m22s$ vs. $215m01s$, 5.0GB vs. 32.4GB) due to dynamic merging and SVD techniques.

**Speedup variants**   Currently, our method supports batch inference for the routing process, while the dynamic merging process handles inputs sequentially. However, it is straightforward to extend our approach to support merging in batches or groups. We can achieve this by first obtaining router weights in batch, then grouping similar data items using the following strategy: (1) Divide into Bins Based on Argmax Indices: First, we divide the data into several bins according to the arg-max indices of the router logits. (2) Cluster Within Each Bin: Then, we cluster (by KMeans) within each bin to group the logits (we set the group number to 20). (3) Average Weights Within Each Group: Within each group, the router weights are averaged to obtain a merged model. Each group corresponds to one merged process, and the group size is typically larger than the batch size, making it very efficient. We have added a group-wise experiment on RoBERTa to illustrate this:

Table 12: Performance of group-wise variant.

| Method | Avg. Normalized Score | Time |
|---|---|---|
| **Task-Arithmetic** | 67.80 | 4m52s |
| **Twin-Merging** | 96.14 | 9m31s |
| **Twin-Merging (group-wise, group number=20)** | 90.02 | 5m14s |

# F    Limitations and Future Work

Our approach shares common limitations with existing merging methods: (1) The underlying theory behind why and when weight interpolation works is not fully understood, though recent works [51, 83] have made interesting observations about weight disentanglement and cross-task linearity. (2) Currently, merging is limited to models with the same architecture and it may be difficult to find a suitable fine-tuned model with specific capacities.

Additionally, while our method focuses on shared and exclusive task-specific knowledge, providing a way to approach fine-tuned model performance and potentially surpass it without additional training, we observe there may be other types of knowledge that remain unexplored: (1) *Evil knowledge*: Useless for any task and distracts the model, obscuring critical knowledge during merging. (2) *Irrelevant knowledge*: Has no impact on merging performance. Our experiments validate the existence of the irrelevant knowledge since we demonstrate that dropping $90\%$ of parameters retains most of the fine-tuned performance, but we have not investigated evil knowledge. Future work may include further investigation and decomposing these different types of knowledge to better ignite the model's full potential without retraining.

# G    Broader Impacts

This paper presents work whose goal is to advance the field of machine learning and model merging research. In terms of positive social impact, twin-merging techniques can achieve multi-task performance of foundation models without retraining expert models, significantly reducing computational and energy costs. Our proposed knowledge modularization and compression techniques make the task-specific enhanced model more accessible and sustainable, paving the way for broader applications and advancements in the field. These techniques effectively align unaligned models by leveraging experts, thus mitigating the harmfulness and biases present in the original models. Additionally, model merging allows the unified model to benefit from the strengths of each task-specific model, even for tasks with private or inaccessible data, enhancing commercial and safety benefits. However, improper merging of biased models may contaminate the merged model. This issue can be addressed by merging a de-bias expert or using sparsity techniques to minimize the impact.

