# OpenReview forum: "Twin-Merging: Dynamic Integration of Modular Expertise in Model Merging"
_NeurIPS.cc/2024/Conference — NeurIPS 2024 poster_

### Official Review · Reviewer_LqLU · 2024-06-27

**Soundness:** 3
**Presentation:** 2
**Contribution:** 3
**Rating:** 6
**Confidence:** 4

**Summary:**

This paper investigates a dynamic and compressive merging method for adapting large-scale models to multiple tasks. The authors claim that adjusting the ratio between shared knowledge and exclusive knowledge is crucial for high-performing model merging, and they devise an algorithm that learns proper coefficients to merge those two kinds of knowledge in a data-dependent manner. They further compress the task-specific exclusive knowledge via SVD to reduce memory storage during merging. They validate their method with language models on discriminative and generative tasks.

**Strengths:**

- Remarkable performance improvement compared with baselines
- The proposed dynamic merging method can be memory-intensive during inference time because it requires all task-specific modules to be stored, but the author mitigates this somewhat via SVD compression. The combination of dynamic merging and compressive merging is very impressive, which induces better performance while considering practical usefulness

**Weaknesses:**

- Lack of technical details on the most important part of the methodology
    - The router module plays a crucial rule that produces input-dependent merging coefficients during inference time. However, the authors do not provide any technical details on the training of the router module (including appendix).
        - Model architecture and training configurations should be provided.
    - Moreover, a description of the validation set the authors to use for the router training should be provided. The authors only describe that dataset as '**_a small validation set_**'. Does the validation set consist of the integration of downstream tasks? or general text corpus? and how the construction of the validation set affects the final merged model performance.
        - I believe details on the construction and amount of samples for the validation set should be presented.
- Limited scope of validation
    - While they evaluate their method with fully fine-tuned models on discriminative tasks, they only validate it on the parameter-efficient fine-tuning regime for the generative tasks. As one of the proposed method's main contributions is parameter compression, it would be more effective when they show the applicability of their method on fully fine-tuned model merging with decoder-only LM or encoder-decoder LM.
- Concerns about scalability
    - While the compressive merging reduces the requirements of memory storage during the inference phase, I speculate that the amount of computation for the merging operation is still a huge burden. The input-dependent dynamic merging requires computations for merging operation per every sample, and I wonder about the **runtime and computation cost during inference phase when the fully fine-tuned models are the target of validation** rather than the parameter-efficient fine-tuning regime in Table 7.

**Questions:**

- details on router network architecture and its training (refer to weakness section)
- details on the validation set construction (refer to weakness section)
---
If there is a misunderstanding from me, please don't hesitate to refute it. I would be happy to discuss this further.

**Limitations:**

Lack of scalability to larger models due to increasing inference cost of instance-dependent dynamic merging.

---

> ### Author Rebuttal · Authors · 2024-08-07
>
> # Response to Reviewer `LqLU`
>
> > Q1. Lack of technical details on the router (Model architecture and training configurations ), validation set for router training. Does the validation set consist of the integration of downstream tasks? or general text corpus?
>
> Thanks for your suggestion, we will revise our paper in the next version.
> Our router is implemented as a three-layer linear network with Leaky ReLU activations and batch normalization.
> We train the router on the validation dataset with a learning rate of 5e-4 for 10 epochs.
>
> The validation set consists of the integration of in-domain downstream tasks, not the general text corpus.  The validation set is taken from a split of the training set, and we use at most 1,000 items for router training for each task.
>
> > Q2. how the construction of the validation set affects the final merged model performance.
>
> As for how the construction of the validation set affects the final merged model performance, if the dataset is too small or imbalanced, it can affect the router's precision and degrade the merging performance. Therefore, we typically ensure the sample numbers from different tasks are the same. Practically, using 1,000 items  (which is about 2% of the discriminative task test sets) is sufficient to achieve comparable performances.
>
> > Q3. They only validate it on the parameter-efficient fine-tuning regime for the generative tasks. They should show the applicability of their method on fully fine-tuned model merging.
>
> We primarily merge using LoRA for the Qwen-14B models because fully finetuning them to obtain task-specific experts would require a huge amount of resources and computation (finetuning the full 14B model requires at least 8 A100 GPUs).
> However, to further demonstrate, we provide experiments on fully fine-tuned LLaMA 7B models for generative tasks (gsm8k and truthfulqa), where our approach still exhibits superior performance.
>
> | Method |  avg. normalized score | Inference Time (/1000 items)|
> |-|-|-|
> | Task-Arithmetic  | 69.89  |  186s  |
> | Twin-Merging  | 88.18  |  198s  |
>
> We have also verified the effectiveness of our method on fully fine-tuned RoBERTa (as shown in Table 2) and ViT-B/32 models (in the global rebuttal section).
>
> > Q4. I speculate that the amount of computation for the merging operation is still a huge burden. The input-dependent dynamic merging requires computations for merging operation per every sample. I wonder about the runtime and computation cost during the inference phase when the fully fine-tuned models are the target of validation rather than the parameter-efficient fine-tuning regime in Table 7.
>
> Please refer to the analysis in the "Inference Efficiency" section of the global rebuttal. Our approach introduces negligible cost in typical generation scenarios and can be easily optimized by group-wise merging.
>
> As for runtime and computation costs for fully fine-tuned models, please refer to the LLaMA 7B time cost in the response to Q3 (186s->198s) and the ViT-B/32 results in the global rebuttal (47m22s vs 215m01s).
> From these tables, we can see that our method introduces only an extra 0.01 seconds per sample for LLaMA7B and is faster than methods like AdaMerging and Surgery, which also improve performance through additional training and modules.

---

> ### Comment · Reviewer_LqLU · 2024-08-09
>
> I appreciate the authors' kind response! Some of my concerns are addressed.
>
> * I still think that TwinMerging's inference time overhead compared with the vanilla task arithmetic in Table 2 of global response is significant (about two times), given that group-wise extension is not included in the reviewed manuscript.
> * Moreover, as from your answer, TwinMerging requires labeled samples from ALL test domains in advance (even though it is validation splits), which is an optimistic assumption compared with AdaMerging, which only requires unlabeled test domain data.

---

> > ### Author Response · Authors · 2024-08-09
> >
> > We thank the reviewer for the feedback. We will address the additional concerns below.
> >
> > > Q1
> >
> > Our original approach significantly outperforms both AdaMerging and Surgery in terms of speed and performance.
> > As evidenced by 1st table from the global rebuttal, our method completes in just 47m22s, compared to 185m35s / 215m01s for AdaMerging/Surgery.
> > Despite the substantial time savings, our approach also delivers superior performance, achieving a score of 95.33, versus 88.50 / 94.40  for AdaMerging / Surgery,
> > **the latter methods invest over 10 times the effort to marginally improve performance**.
> >
> > Furthermore, our group-wise variant that matches the efficiency of Task-Arithmetic (5m14s vs 4m52s) still holds superior performance (92.02 vs 67.80). We will include this variant in the revised version of our paper.
> >
> > > Q2
> >
> > To clarify, AdaMerging requires **the actual test sets** (as confirmed in Section 3.2.2 of the original paper, or the code from `src/datasets/common.py` line 142 in the original codebase). This means that, for the example in Table 5, for the unseen domain dataset,
> > AdaMerging theoretically requires access to these unseen domain test sets to approximate the optimal model, though without true labels.
> > This poses a significant challenge for online deployment scenarios, where test inputs are unpredictable and streaming.
> > Even if we can access to the test set in in offline scenarios, scaling the test set to a large number, such as 1,000, makes the process highly inefficient.
> >
> > Additionally, AdaMerging relies on entropy optimization for unsupervised training, making it **applicable only to classification tasks**.
> > Given the growing dominance of large **generative** foundation models like LLaMA, GPT-4, and diffusion models,  AdaMerging’s focus on classification limits its scalability and applicability.
> >
> > In contrast, our approach only requires in-domain validation data corresponding to each expert. For instance, in Table 5, we used 5 experts to handle 3 unseen datasets, necessitating only the validation sets corresponding 5 experts.
> > This is because the validation dataset is meant to enable the router to learn and identify specific exclusive knowledge in domain.
> > Please notice that this training process is **agnostic to the actual test distribution**.  Whether the test distribution scales to 10, 100, or even 1000 tasks, we still only need those initial 5 validation datasets.
> > Our method, however, is versatile and can handle any type of task, including generative tasks with Qwen-14B in Table 2, and scales up to **72B** on generative tasks (Table 3), offering broader applicability and alignment with current AI trends.

---

> ### Comment · Reviewer_LqLU · 2024-08-10
>
> Thanks for your detailed rebuttal.
>
> Authors' claims are convincing and I will raise my score 5 -> 6

---

> > ### Comment · Reviewer_LqLU · 2024-08-11
> >
> > Sorry for my reverse (6 -> 5), but I would like to discuss my second worry with the authors further.
> >
> > After reviewing the authors' responses, I still lean toward the negative side due to their reliance on the validation set.
> > While the method is agnostic to the test distribution, its effectiveness is unclear under severe distribution shifts in image domains (that AdaMerging focuses on) compared to distribution shifts in language domains. That is, I still doubt the performance sensitivity against distribution shifts and varying amounts of validation set.
> >
> > Moreover, in the real-world deployment scenario, I think it is much harder to gather validation sets for each merging ingredient fine-tuned model from different institutions (due to commercial / privacy issues) compared with test-time unlabeled incoming samples.
> >
> > Given that, my concerns about Twin-Merging's reliance on ID validation still seem to be weaknesses, even though it boosts performance significantly.

---

> > > ### Author Response · Authors · 2024-08-12
> > >
> > > Thanks for your feedback. We will address the concerns below.
> > >
> > > > Q1: image domain distribution shift & varying amounts of validation set
> > >
> > > We want to clarify that **our method addresses distribution shifts through dynamic merging, which adapts to test inputs**, as shown in our NLP results (Table 5) and Section 4.5, although the preprocessing stage (router training/knowledge modularization) is indeed agnostic to the test distribution.
> > >
> > > To address your specific concern, we conduct additional experiments on image domain shifts using Gaussian noise corruption. The results demonstrate that Twin-Merging is robust against image domain distribution shift.
> > >
> > > | Method | Avg. Before Corruption | Avg. After Corruption |
> > > |-|-|-|
> > > | Task-Arithmetic  | 84.3  |  65.6  |
> > > | AdaMering  |  91.7 |  73.4  |
> > > | **Twin-Merging**  | **96.9** |  **79.2**  |
> > >
> > > In terms of varying amounts of validation set, we actually have a related experiment in Figure 4, where the router validation set is varied from 2,000 to 7,000, and we can observe consistent superior performance.
> > > To further demonstrate, We also add experiments with reducing the validation data number to 100 per task for generative tasks, which is 1/10 of the original size, and we observe that the performance does not change significantly:
> > >
> > > | Method | val-1000 | val-100 |
> > > |-|-|-|
> > > | Twin-Merging  | 102.38  |  101.23 |
> > >
> > > > Q2: harder to gather validation sets for each merging ingredient fine-tuned model from different institutions (due to commercial / privacy issues)
> > >
> > > Firstly, to clarify, **our approach does not strictly require the validation set to be taken from each in-domain dataset used by the experts**. For example, we can utilize the opensource Dolly dataset to represent the MMLU expert (L619-L620). Furthermore, **we do not need to gather validation datasets for all experts**. As illustrated in Table 5, our approach still works effectively without gathering specific validation datasets for QNLI, MNLI, RTE, and MMLU.
> > >
> > > In practice, we can select a subset of accessible experts and gather representative validation data for them, which is typically not difficult.
> > > This actually stems from the key assumption of our approach, that **in-domain knowledge contains complementary elements that can effectively address out-of-domain inputs when combined properly [1,2]**. By dynamically inferring optimal weights to combine this modularized knowledge based on the test input, our method offers better collective generalization against the unpredictable heterogeneous input.
> > > In contrast, AdaMerging may limits in imprecise entropy approximation and the lack of supervised guidance.
> > > To better demonstrate the advantage of our approach, we conducted additional generalization experiments using the same settings as in the AdaMerging paper:
> > >
> > > | Method | EuroSAT | MNIST | Avg. Acc |
> > > |-|-|-|-|
> > > | AdaMering  | 45.9  | 90.1 | 68.0  |
> > > | **Twin-Mering**  | **53.2** | **92.9** | **73.5** |
> > >
> > > Secondly, we want to emphasize that our approach is primarily designed for **real-world LLM online serving**, where models are deployed with a continuous stream of inputs without gradient updates, aligning with trends in LLM deployment [3,4].
> > > - **Unpredictable, Heterogeneous Data Streams**: The nature of these data streams means that traditional batch techniques are inefficient, and any single expert is insufficient. This is why we introduce our dynamic merging technique.
> > > - **Latency and Storage Considerations**: To address critical concerns around latency and storage, we shift time-consuming router training and knowledge modularization to the preprocessing stage. Additionally, we employ SVD techniques to reduce storage requirements. A detailed analysis of the FLOPs for these serving scenarios is provided in the global rebuttal.
> > > - **Broad Applicability**: Our method applies to both NLP and CV domains, spanning discriminative and generative tasks, making it highly adaptable for deploying and scaling AI models.
> > >
> > > In contrast, the AdaMerging post-training technique incurs significant latency due to its reliance on test-time training, making it more suitable for offline evaluations rather than real-time LLM deployment.
> > >
> > > [1] Fusing Models with Complementary Expertise [ICLR24]
> > >
> > > [2] Knowledge Fusion of Large Language Models [ICLR24]
> > >
> > > [3] Mooncake: Kimi's KVCache-centric Architecture for LLM Serving
> > >
> > > [4] LLM Inference Serving: Survey of Recent Advances and Opportunities

---

> > > > ### Author Response · Authors · 2024-08-13
> > > > **Sincerely looking forward to your feedbacks**
> > > >
> > > > We sincerely appreciate your valuable feedback and concerns regarding the clarity of our descriptions and the validation sets.
> > > >
> > > > We hope our response has effectively addressed all your concerns. Your insights are crucial for improving our work, and we are open to further discussion if you have any questions about our response.
> > > >
> > > > With the effectiveness in merging performance (even outperforming the oracle at times), efficient storage, minimal time cost, and a reasonable assumption regarding the validation dataset—as recognized by Reviewer `2gMR` and `Q3uE`—we believe that our approach will become increasingly practical and significant in the era of large language models. We hope that these insights and outcomes can contribute to the community. We appreciate your time and would be very grateful if you could re-evaluate the paper’s rating.

---

> ### Comment · Reviewer_LqLU · 2024-08-14
>
> I want to express my severe gratitude for the authors' kind response and my regret for my unprofessionalizm, such as the score-reversing behavior.
>
> Thanks to the discussion with the authors, I could also extend my sight, not only gain a clearer understanding of the paper.
>
> I will raise my rating accordingly.

---

> > ### Author Response · Authors · 2024-08-14
> > **Thank you**
> >
> > We appreciate the reviewer's recognition of our work's effectiveness and the score increase. We will revise our work to include clearer illustrations. Thank you for your time and valuable suggestions!

---

### Official Review · Reviewer_2gMR · 2024-07-10

**Soundness:** 3
**Presentation:** 4
**Contribution:** 3
**Rating:** 6
**Confidence:** 4

**Summary:**

This paper attempts to resolve an issue of destructive interference with model merging techniques. It proposes to maintain a shared base model and separate task-specific knowledge structures that can dynamically be combined at test time.
The paper presents some a nice to buttress the drive home their methodology

**Strengths:**

- Code provided
- Initial analyses of the existence of interference even when separate parameter sets are fine-tuned with LoRA is interesting though a bit obvious in hindsight be useful to empirically validate
- Initial experiments validate the hypotheses of interference and shared / exclusive knowledge
- Reasonably thorough experimentation

**Weaknesses:**

1. My primary issue with the paper is why this is a viable alternative to just keeping the base model back-bone and LoRA low-rank vectors of similar rank to the $v_t$s (Algo 1) (and instantiating the task specific model on the fly).  Twin-Merge has memory overhead (v_t for T tasks) which brings it more into the realm of “saving task-wise adapters/LoRA” whose size grows as the number of tasks. It is unclear from the existing set of experiments why a practitioner would use this method over keeping separate task low-rank (small memory footprint) adapters and just using these at test time (either statically per task or dynamically — https://arxiv.org/pdf/2306.14870 ) ?
    1. I understand that the paper performs a primary comparison to other model merging methods like DARE and Task Arithmetic but these methods have zero extra memory overhead (after the model is merged) — they don’t enjoy the expressivity that test time adaptation + extra memory gives them.
2.  Claims of generalization to unseen tasks might not be valid. Specifically, it is interesting to see that the boost in performance for Twin Merging in Table 2 is much more significant than for Table 5 with unseen generalization. This makes me wonder if the deltas in table 5 are actually significant.


I am open to raising my score if the authors can convince me of the utility of the method -- in light of [1] above.

**Questions:**

* How are the $v_t$s represented / stored (Algorithm 1) ?
    * Are the v_ts stored as the low rank matrices that are later multiplied to obtain the appropriate matrix size ?  As written, it seems like each $v_t$ is of size equal to the size of the full parameter space and so ends up as a memory burden.
* For Figure 4 (right), is the fine-tuned storage size much larger because you are doing full fine-tuning instead of LoRA fine-tuning ?
    * If you are doing LoRA fine-tuning is the gap because the rank of the lora matrices are larger >> than the rank of the final $v_t$s above ?
* Maybe I missed this but where is the “(Best Storage)” option for Twin Merging in Table 2  described ?
* For Table 4, would it be possible to provide the results for Twin-Merging only before showing the additive results ? It’s hard to make out whether the other methods are contributing anything at all

**Limitations:**

Broader impact statements and limitations are discussed in the paper

---

> ### Author Rebuttal · Authors · 2024-08-07
>
> # Response to Reviewer `2gMR`
>
> > Q1. Why this is a viable alternative to just keeping the base model and LoRA low-rank vectors? Why we would use this method over keeping separate task low-rank adapters and just using these at test time ?
>
> Table 2 has proven that our knowledge modularization technique outperforms directly using task-specific LoRA at test time in generative tasks, exhibiting an average score of 102%.
> Compared to several common strategies to utilize task-specific adapters, Twin Merging is still a promising method to improve performance:
> -  **Directly Route to Top-1 Expert**, which is equivalent to the fine-tuned baseline in Table 2: This is the simplest but highly impractical approach, as it relies heavily on the router. It has the worst expected performance for out-of-domain data when the router cannot properly predict. **Twin Merging outperforms this method (102.38 > 100.0) and has superior performance on unseen tasks (Table 5)**, as it can benefit from complementary knowledge from different exclusive sources. Additionally, routing to the top-1 expert requires storing all experts, which can be a large storage requirement when used with fully fine-tuned models.
> -  **Combining Multiple Experts**: Without the isolation of shared and exclusive knowledge, combining multiple experts suffers from interference problems, as the redundancy of shared knowledge may obscure the key knowledge required for tasks, and conflicts between the exclusive knowledge are also unavoidable (analyzed in Section 3.2).
>     - *Statically Combining*: This is equivalent to "Task Arithmetic" when using LoRA. **Twin Merging outperforms static combination (102.38 > 96.61)**.
>     - *Dynamically Combining*: This is equivalent to the method (A) in the following table. **Twin Merging(B) outperforms dynamic combination (102.38 > 97.03)**.
>
> |Method|RoBERTa|Qwen|
> |-|-|-|
> |Pretrain+Dynamic Merging(A)|85.90|97.03|
> |Shared+Dynamic Merging(TwinMerging,B)|96.14|102.38|
>
> > Q2. Baselines like DARE and Task Arithmetic don’t enjoy the expressivity that test time adaptation + extra memory gives them.
>
> We add baselines like AdaMerging and Representation-Surgery that have extra time cost and memory consumption in the "More Baseline" section of the global rebuttal.
> **While consuming less time cost and memory consumption, our approach superior in performance (95.33 > 94.04 > 88.50)**.
> As analyzed in the "Inference Efficiency" section of the global rebuttal, our method actually introduces neglectable cost compared to the total generation (0.039s per sample in Table 7), and can be further optimized by the group-wise merging.
>
> > Q3. Twin Merging Performance in Table 2 is much more significant than in Table 5 with unseen generalization.
>
> This is because we present **unnormalized scores** in Table 5. We cannot directly normalize them as we do not have the corresponding expert on unseen datasets to get upper-bound performances (as noted in line L242).
> This leads to relatively lower scores due to the narrower score ranges for tasks like RTE (max 66.43 vs max 91.71 for QNLI) and MMLU (max 68.03).
>
> If we use the maximum score from Table 2 as the upper bound for normalization,
> we observe more significant improvement for Table 5, e.g. 91.16 -> 96.98 for MMLU (We present the "unstrictly-normalized" scores in parentheses):
>
> |Method|QNLI+MNLI+RTE|MMLU|
> |-|-|-|
> |MTL|44.63(55.87)|63.74(93.69)|
> |Task-Arithmetic|53.92(67.42)|62.02(91.16)|
> |Twin-Merging|55.86(71.92)|65.98(96.98)|
>
> The scores are lower than in Table 2 because the in-domain knowledge may be unrelated or even harmful to the test data, leading to a performance downgrade.
> However, our approach helps mitigate this effect by dynamically adjusting the merging weight.
>
> > Q4. How are the $v_t$ represented/stored? Are the $v_t$ stored as the low rank matrices? It seems size of $v_t$ is equal to the size of the full parameter space.
>
> We do not directly store $v_t$ since it is the same size as the original parameter. As detailed in Appendix D5, we further compress $v_t$:
> Given the size-$m$ decomposition $v_t = \mathbf{U}_t \mathbf{\Sigma}_t \mathbf{V}_t^T$, we select the top-$r$ singular values to form $\mathbf{U}_t(r) \mathbf{\Sigma}_t(r) \mathbf{V}_t(r)$. **We store only $\mathbf{U}_t(r)$, $\mathbf{\Sigma}_t(r)$, and $\mathbf{V}_t(r)$**.
>
> During merging, we decompress these matrices by extending $\mathbf{U}_t(r)$, $\mathbf{\Sigma}_t(r)$, and $\mathbf{V}_t(r)$ to size-$m$ by filling with zeros, allowing us to recover $v_t$ via their product. This operation is only at the matrix level; once we obtain the merged matrix, we discard the decompressed matrices, ensuring efficient storage.
>
> > Q5. For Figure 4, is the fine-tuned storage size much larger because you are doing full fine-tuning ? If you are doing LoRA fine-tuning is the gap because the rank of the lora matrices are larger than the rank of the final $v_t$ ?
>
> Yes, we mainly show the full fine-tuning results for RoBERTa in Figure 4.
> As analyzed in Appendix E, for LoRA fine-tuning, the typical rank is 32, but in our experiments, we can use a rank-1 for the best storage efficiency and still obtain good results (refer to Table 9), which is much smaller in storage.
>
> > Q6. Where is the "Best Storage" option
>
> The "Best Storage" option refers to the rank-1 compression via SVD, as illustrated in Table 8 and Table 9. The detailed storage analysis is provided in Appendix E.
>
> > Q7. Showing the additive results for the Ablation Study
>
> Thanks for your suggestion. We show the additive style ablation as follows:
>
> |Method|RoBERTa|Qwen|
> |-|-|-|
> |Pretrain|41.69|91.06|
> |Shared|67.80|96.61|
> |Dynamic Merging|81.47|87.77|
> |**Shared+Dynamic Merging(Twin Merging)**|**96.14**|**102.38**|

---

> > ### Comment · Reviewer_2gMR · 2024-08-10
> > **Response**
> >
> > Hi authors,
> > Thanks for your response. And including the additional baselines.
> > > Directly Route to Top-1 Expert
> >
> > I'm wondering why you decided to do only Top-1 expert in this baselines instead of soft router weights like you do.
> > I think a better comparison would have been to have Top-K or even just a soft routing like you guys have in the paper. This would then be a better way of demonstrating that Twin merge is superior. As it stands I don't think there is sufficient evidence to say your method is better than the simple LoRA adapter router baseline -- esp given the relatively small delta (102.38 > 100.0) (how does this breakdown to individual task scores btw ?)
> >
> > Based on the explanations and updated experiments. I'm raising my score -- but it would be great if the more extensive version of the experiment above (as I mentioned, is included in the paper)

---

> > > ### Author Response · Authors · 2024-08-10
> > >
> > > We thank the reviewer for raising the score! We will address the additional comments below.
> > >
> > > > Q1. why listed Top-1 expert as baseline in original paper, not the "soft merging" method
> > >
> > > We choose the "Route to Top-1 Expert" as an **oracle** baseline to highlight the performance gap between common merging techniques and **the ideal scenario**.
> > > Because this baseline typically performs the best [1], as shown in the table below (taken from the response to Q1):
> > >
> > > | Method |  RoBERTa | Qwen|
> > > |-|-|-|
> > > | Top-1 (oracle)  | 100.00  |  100.00  |
> > > | Pretrain+Dynamic Merging(A)  | 85.90  |  97.03  |
> > > | Shared+Dynamic Merging(TwinMerging,B)  | 96.14 |  102.38  |
> > >
> > > The "soft merging" (A) performs worse than the oracle, due to the interference between the different models, which is consistent with the findings in [1].
> > > However,Twin-Merging mitigates it by modularizing shared and exclusive knowledge, leading to improved performance and, in some cases, even surpassing the oracle baseline (102.38 > 100.00).
> > >
> > > We appreciate the reviewer's suggestion to include the "soft merging" as a baseline for better clarity. We will incorporate this into our revised version of the paper.
> > >
> > > > Q2. unsufficient evidence to say your method is better than the simple LoRA adapter router baseline -- esp given the relatively small delta (102.38 > 100.0) (how does this breakdown to individual task scores btw ?)
> > >
> > > We’ve provided a detailed breakdown of the scores in Table 9, Appendix D7.
> > > It’s important to note that the LoRA router baseline represents the **oracle performance**, which typically achieves the best results, as explained in the above.
> > > Surpassing this baseline is extremely challenging.
> > > However, our method achieves nearly identical performance ( 99.87 vs. 100 on MMLU) or even outperforms the oracle in some datasets (over a 14% improvement on CNN/DM).
> > >
> > > Additionally, a key limitation of the simple LoRA router baseline is its difficulty in adapting to unseen tasks, as the router often struggles to predict the correct top-1 result.
> > > In contrast, our approach demonstrates better performance on unseen tasks (as shown in Table 5), leveraging complementary knowledge from different exclusive sources to enhance collective intelligence.
> > >
> > > [1] Exploring the Benefits of Training Expert Language Models over Instruction Tuning [ICML23]

---

### Official Review · Reviewer_Q3uE · 2024-07-10

**Soundness:** 2
**Presentation:** 3
**Contribution:** 2
**Rating:** 6
**Confidence:** 3

**Summary:**

`Twin-Merging` proposes a method for task merging which tackles two issues:
  * Task interference: The proposed `Twin-Merging` explicitly model shared vs task-specific knowledge to potentially reduces redundancies across the task vectors, which may lead to subpar task merging results
  * Dynamic merging: Usually, task merging weights are only task dependent and determined only once. In contrast, here, the weights are determined at the input level using a router akin to Mixture of Experts design.

**Strengths:**

* Compressing the task vectors using SVD is beneficial for memory usage
* Performance improves over standard  task merging approaches
* Experiments are also conducted on large scale models (72B)

**Weaknesses:**

* **The per-input routing design seems highly impractical**.  If I understood correctly, the task merging weights are compute on-the-fly for each individual input. This means that:
	  * The design does not support batching as we need a different merged model for each input in the batch
	  * Every time a new input comes, we need to first get the merging weights by executing the fuser, then uncompress the task vectors, build the merged model, and finally run the inputs through it. This does not seem very hardware friendly, even if the task vectors are low-rank compressed using SVD. In contrast, in standard task merging, we only ship one merged model.
	  * Because the fuser $\mathcal{R}$ takes as inputs the  *last-layer token embeddings from the shared expert* (line 186), does it mean that every input require two forward passes (one through the fuser, then one through the merged model) ?
- The paper does not really convey the importance/novelty of **knowledge modularization** strongly enough. In essence, the basic task arithmetic (**A**) already performs some form of modularization where the pretrained model is the *shared knowledge* and task finetuned models (task vectors) are *task specific*. In contrast, knowledge modularization(**B**) **redefined the task vectors** relatively to the merged model (rather than the pretrained one) and compress them via SVD.  In my opinion, comparing **A** and **B** would be a better ablation experiment than the one in Table 6 where the merged expert is replaced by a random specific one.

- **Baseline**: Since the proposed design uses additional validation data (to finetune the router), it would be fair to also compare to the more recent/stronger baseline `AdaMerging: # Adaptive Model Merging for Multi-Task Learning` (Yang et al), which also assumes extra data available.

**Questions:**

* I do not fully understand the conclusion of **Section 3.1**:
	  * the assumption of `Ties Merging` is that redundant parameters can be hard to merge. But they do not really make assumptions on whether these parameters were trained jointly/with overlap > therefore I'm not sure how the LoRA experiments contradicts this insight: Even if the LoRA modules are trained separately, they may still have redundant statistics at the end of training ?
	  * Similarly, the notion of *similar tasks* or *task interference* is hard to define in general.  So it is not clear to me how significant the results of the XSUM/DailyMail experiment are.
  * Based on the result of table 4.5 it looks like Twin Merging does not often behave as a model merging method ? It seems that most of the time the samples are routed to their respective task-specific experts. If that insight is true, it is not too surprising that `TwinMerging` performs on-par with the finetuned baseline.
  * In Table 7: Shouldn't `Model Merging` and `Twin Merging` also integrates the cost of training the initial task vectors in **training cost** ? Otherwise it seems unfair to the MTL baseline.

**Limitations:**

The paper discusses limitations inherent to task merging in appendix F. However, as discussed in the Weaknesses section.

---

> ### Author Rebuttal · Authors · 2024-08-07
>
> # Response to Reviewer `Q3uE`
>
> > Q1. The design does not support batching
>
> While the router process supports batching, the dynamic merging process currently handles inputs sequentially. However, **it is straightforward to extend the merging process to support batching**. As detailed in the "Inference Efficiency" section of our global rebuttal, we can achieve this by clustering and rearranging data items based on the router logits into groups. This approach significantly reduces the number of merging operations required, with minimal impact on performance. By doing so, we maintain the benefits of our approach while efficiently processing inputs in batches.
>
> > Q2. Does every input requires two forward passes (through the fuser and through the merged model) ?
>
> For the fuser, yes, it requires one forward pass to induce the merging weights. However, the merging model typically requires hundreds of forward passes for generation (e.g., 300 tokens for summarization). Therefore, the additional cost is typically negligible, referring to analysis in the "Inference Efficiency" section of the global rebuttal.
>
> > Q3. The basic task arithmetic (A) already performs some form of modularization where the pretrained model is the shared knowledge and task finetuned models are task-specific. Knowledge modularization(B) redefined the task vectors relative to the merged model and compressed them via SVD. Comparing A and B would be a better ablation experiment than the one in Table 6.
>
> Thank you for your suggestion. We have revised the ablation study table as shown below:
>
> |Method|RoBERTa|Qwen|
> |-|-|-|
> |Pretrain+Dynamic Merging(A)|85.90|97.03|
> |**Shared+Dynamic Merging(TwinMerging,B)**|**96.14**|**102.38**|
>
> We observe that A performs worse than B (85.90 vs 96.14), which can be attributed to two main reasons:
> - The pretrained model may contain relatively sparse shared knowledge that benefits the input tasks. In contrast, the shared expert, constructed by merging task-specific experts, contains more abundant and diverse shared knowledge.
> - Modulizing knowledge by subtracting the pretrained model does not effectively mitigate interference, as it does not consider exclusive knowledge specific to each task. This explains the performance gap between the task vectors and the fine-tuned experts, as analyzed in Section 3.2.
>
> > Q4. Compare to AdaMerging which assumes extra data is available.
>
> Thank you for your suggestion.
> We add AdaMerging which need extra validation data in the "More Baseline" Section of the global rebuttal.
> Our approach still outperforms them with less time costs (95.33>88.50).
>
> > Q5. They do not really make assumptions on whether these parameters were trained jointly/with overlap. therefore I'm not sure how the LoRA experiments contradicts this insight: Even if the LoRA modules are trained separately, they may still have redundant statistics at the end of training ?
>
> To clarify, our study focuses on the "parameter interference" phenomenon as defined by the Ties-Merging paper[1]. This refers to the conflict of parameters **at the same position across task experts**, e.g., the sign disagreements of the up-projection layer in different task models, which is the main focus of the Ties-Merging method.
> Our Section 3.1 experiment demonstrates that even when task-specific modules are trained without overlap thus merging without overlap, interference still occurs.
> This indicates that Ties-Merging approach does not fully resolve parameter interference.
>
> > Q6. The notion of similar tasks/task interference is hard to define in general. So it is not clear to me how significant the results of the XSUM/DailyMail experiment are.
>
> We use "task interference" from MTL literature [2] to describe the distinct nature of different task types. For instance, summarization, math reasoning, and code generation each require different forms of responses. Conversely, XSUM and DailyMail are both summarization tasks, handled similarly by the model. Our experiments showed that even similar task types experience interference, prompting us to explore finer-grained relationships between tasks, such as the knowledge types (Sec 3.2).
>
> > Q7. It seems that most of the time the samples are routed to respective task-specific experts. If that insight is true, it is not too surprising that TwinMerging performs on-par with the finetuned baseline.
>
> To clarify, we **do not directly route samples to task-specific experts** (equivalent to the fine-tuned baseline in Table 2), which is highly impractical when facing unknown test distributions.
> Instead, we combine shared and exclusive knowledge, which leads to **even better performance sometimes**, e.g., averaging 102% over the fine-tuned baseline on generative tasks (Table 2) and 101% on COLA (Table 8), and better unseen generalization (Table 5).
> By isolating different types of knowledge and composing them dynamically, we avoid redundancy and leverage complementary information from both shared and exclusive sources, resulting in improved performance.
>
> > Q8. Shouldn't Model Merging and Twin Merging also integrates the cost of training the initial task vectors in training cost ? Otherwise it seems unfair to the MTL baseline.
>
> We did not include training time in Table 7 because merging methods can directly download task experts from Hugging Face/PyTorch Hub without post-training. However, to demonstrate, the training time and cost for custom fine-tuning from scratch are as follows:
>
> |Method|TrainingTokens|TrainingCost|Performance|
> |-|-|-|-|
> |MTL|536.35M|10h32min|94.31|
> |Task-Arithmetic|536.35M|10h32min|96.61|
> |Twin-Merging|536.92M|10h35min|102.38|
>
> Our approach shows an 8.5% performance improvement over MTL with only a 0.1% increase in training tokens and a 0.4% increase in training time.
>
> [1] TIES-Merging: Resolving Interference When Merging Models [NIPS23]
>
> [2] Mitigating Task Interference in Multi-Task Learning via Explicit Task Routing with Non-Learnable Primitives [CVPR23]

---

> > ### Comment · Reviewer_Q3uE · 2024-08-11
> >
> > Dear authors,
> >
> > thanks for your response and clarifications:
> >   * The new ablation on A vs B is more convincing in showing the benefit of explicitly building shared/specific task vectors
> >   * The group-wise variant of Twin-Merging is interesting, and it's good to see that performance of the method does not drop significantly in that case. However I do wonder how it would perform
> >
> > Since my main concerns have been addressed, I will raise my score to weak accept: Overall I think the paper is technically solid, and I appreciate the authors' effort in clearly portraying the efficiency/memory cost of the method. However, I also think that the paper introduces several new assumptions departing from traditional task merging (labeled data + extra parameters/router model + per-input dynamic behaviour requiring extra processing for batched inference), and the writing would further benefit from making these differences clearer for fair comparison (e.g. it would be more fair to make Adamerging the standard baseline in the experiments section since it also requires extra but unlabelled data in contrast to Ties-Merging).
> >
> > **Note:** ~[NIPS23]~ -> [NeurIPS23]

---

> > > ### Author Response · Authors · 2024-08-13
> > >
> > > We thank the reviewer for their feedback and for raising the score! We will address the additional comments below.
> > >
> > > > Q1. I wonder how group-wise merging would perform
> > >
> > > The insight is that the router logits indicate the relevance of different exclusive knowledge modules to specific input samples.
> > > Inputs have similar router logits require similar knowledge, i.e., similar merging models.
> > > To leverage this similarity, inputs can be grouped based on their router logits. Within these groups, the merging weights are expected to be similar and can be approximated by an averaged representation.
> > >
> > > To begin, you can divide the inputs into bins based on the arg-max indices of the router logits, which represents the most relevant domain or knowledge module for each input.
> > > To further refine into groups, apply K-means clustering within each bin directly on the router weights.
> > > Once the clustering is done, average the router weights within each group. The model is then merged based on these averaged router weights, allowing the inference for the entire group to be performed using a single model.
> > >
> > > > Q2. the paper introduces several new assumptions departing from traditional task merging (labeled data + extra parameters/router model + per-input dynamic behaviour requiring extra processing for batched inference)
> > >
> > > To clarify, we want to highlight that these assumptions are not entirely new but build upon those in previous works such as FisherMerging (NeurIPS22), DARE (ICML24), and Surgery (ICML24). FisherMerging, for instance, utilizes a validation dataset to adjust merging weights. DARE introduces pre-merging techniques like Sparsify to enhance performance, while AdaMerging and Surgery focus more on post-merging techniques.
> > > Specifically, AdaMerging assumes access to an offline test set and dynamically adapts to it by introducing additional coefficients at every layer, conducting unsupervised training across multiple iterations on the test set (without labels) to refine the model. Surgery goes even further by assuming that test data IDs are accessible during inference, allowing it to insert corresponding task-specific adapters to leverage task-specific knowledge.
> > >
> > > In contrast, our key insight is that **in-domain knowledge, when combined appropriately, can effectively address out-of-domain inputs**, eliminating the need for offline test dataset access or test ID information.
> > > To achieve this, we significantly reduce the additional parameters required, moving from several task-specific adapters across the entire model to highly sparsed exclusive knowledge representation and a single, simple MLP that infers optimal weights for combining modularized knowledge based on the test input. We train this MLP using a small validation dataset rather than an unlabeled test set, which is more suitable for LLM serving, a current trend in AI.
> > >
> > > It is important to emphasize that, like previous methods, our approach only uses **a single merged model** to actually perform the task during the inference phase.
> > >
> > > In summary, while our method introduces techniques across preprocessing, additional parameters, and post-merging stages—similar to previous methods—it is distinct in its approach and insight. This distinction is validated by our experimental results, which demonstrate superior performance with only 1/4 the effort of AdaMerging and 1/5 the effort of Surgery (47m22s vs. 185m35s/215m01s) and just 15.4% of the storage cost of Surgery (5.0GB vs. 32.4GB). We aim to further refine these assumptions in future work.
> > >
> > > > Q3. the writing would further benefit from making these differences clearer for fair comparison
> > >
> > > We thank the reviewer for the suggestion. We will further elaborate on the differences between previous merging methods and incorporate the AdaMerging comparison into our revised version of the paper.

---

### Official Review · Reviewer_vkK5 · 2024-07-22

**Soundness:** 3
**Presentation:** 3
**Contribution:** 3
**Rating:** 5
**Confidence:** 5

**Summary:**

In this paper, the authors introduce the Twin-Merging to merge language models, aiming to close the performance gap between conventional model merging techniques and fine-tuned models, while improving adaptability to data heterogeneity. By modularizing and dynamically merging shared and task-specific knowledge, the authors show that Twin-Merging outperforms existing model-merging methods and approaches the performance of fine-tuned models across various settings and domains.

**Strengths:**

+ The paper is overall well written and easy to follow. The idea of dynamic merging to dynamically merge shared and exclusive knowledge based on the test inputs is interesting.

+ The experiments show clear superiority over prior methods.

**Weaknesses:**

+ The proposed method dynamically merges the models with the varying inputs, which can be extremely time-consuming in practice. It would be better if the authors propose some mechanisms to address this issue.

+ In figure 3, the authors compare 1-model merging with 8-model merging. Why here 8-model merging is used for comparisons? Why not 2-models merging is compared? More explanations should be provided here.

+ Some highly related works are missing in the related work section, such as MuDSC[1]. The differences between these works should be clarified.

[1] Training-Free Pretrained Model Merging, CVPR 2024

[2] REPAIR: REnormalizing Permuted Activations for Interpolation Repair, ICLR 2023.

**Questions:**

Please see the Weaknesses

---

> ### Author Rebuttal · Authors · 2024-08-07
>
> # Response to Reviewer `vkK5`
>
> > Q1. The proposed method dynamically merges the models with the varying inputs, which can be extremely time-consuming in practice.
>
> Please refer to the "Inference Efficiency" section of our global rebuttal. Our method achieves superior performance (95.33 vs 94.04) with less time and storage cost compared to AdaMerging and Surgery baselines  (47m22s vs 215m01s, 5.0GB vs 32.4GB). As shown in Table 7, our approach adds only 0.039 seconds per sample while improving performance by 28.34% for discriminative tasks. Additionally, our method supports common inference speedup techniques and offers efficient group-wise merging variants.
>
> > Q2. In Figure 3, Why 8-model merging is used for comparisons? Why not 2-models merging is compared?
>
> We chose to highlight the 8-model merging scenario to illustrate the degradation gap that can occur in practical scenarios, as typically merging multiple models is more common.
> We actually have demonstrated the performance for 2 to 8 model merging in the left figure of Figure 4.
>
> > Q3. More works (MuDSC and REPAIR) in relative works
>
> Thank you for your suggestion. MuDSC and REPAIR can be categorized as Linear Mode Connectivity (LMC) based methods, which we have discussed in the relative work section (L79-L82).
> MuDSC addresses the issue of inconsistent similarity calculations in activation and weight spaces by designing a merging framework with dual-space constraints to ensure high similarity in both spaces between models.
> REPAIR addresses the problem of collapsed variance in activations during interpolation by rescaling pre-activations, thereby mitigating performance degradation.

---

> > ### Author Response · Authors · 2024-08-14
> >
> > We sincerely appreciate your valuable feedback and concerns regarding the clarity of our descriptions.
> >
> > We hope our response has effectively addressed all your concerns.  Your insights are crucial for improving our work, and we are open to  further discussion if you have any questions about our response.
> >
> > With the effectiveness in merging performance (even outperforming the oracle at times), efficient storage, minimal time cost, and a  reasonable assumption regarding the validation dataset—as recognized by  Reviewer `2gMR` , `Q3uE` and `LqLU`—we believe that our  approach will become increasingly practical and significant in the era  of large language models. We hope that these insights and outcomes can  contribute to the community. We appreciate your time and would be very  grateful if you could re-evaluate the paper’s rating.

---

### Author Rebuttal · Authors · 2024-08-07

# Global Rebuttal

Thank all four reviewers for their constructive feedback which has helped us to improve the clarity and contribution of our work.
The below contains a rebuttal for remarks that are common to most reviewers.

## 1. More Baseline (To `Q3uE`, `2gMR`)

To compare with baselines that require additional datasets and test-time adaptation, we add experiments on ViT-B-32 for 8 CV tasks merging (SUN397 Cars RESISC45 EuroSAT SVHN GTSRB MNIST DTD) following the AdaMerging paper [1] and Surgery [2].

| Method |  Avg. Normalized Score | Additional Time Cost. | VRAM |
|-|-|-|-|
| **Pretrained**| 52.02 | 18m48s | 3.6GB |
| **Fine-tuned**| 100.00| 18m48s | 28.8GB |
| **Weight Averaging**| 72.30 | 18m50s | 3.6GB |
| **Task Arithmetic** | 76.50 | 21m34s | 3.6GB |
| **Ties-Merging** | 75.10 | 19m24s | 3.6GB |
| **AdaMerging**| 88.50 | 185m35s | 3.6GB |
| **Surgery**| 94.04 | 215m01s | 32.4GB |
| **Twin-Merging (Ours)**| **95.33** | 47m22s | 5.0GB |

We use the best version from AdaMerging and Surgery, and the 90% sparsity for our twin Merging.
AdaMerging introduces task-wise or layer-wise learnable parameters to improve the merging performance,
while Surgery adds post-merging task-specific modules to shift representation towards the input tasks.
They both need to be trained on the eight task val set for a long time.
Moreover, Surgery needs to know the task type before inference and requires eight finetuned models and the merged model forward during the merging process, thus exhibiting large VRAM.
**In contrast, our method can robustly handle heterogeneous test inputs and has very efficient storage, exhibiting minimal time cost.**

## 2. Inference Efficiency (To `vkK5`, `Q3uE`, `2gMR`, `LqLU`)

- Currently, our method supports batch inference for the routing process, while the dynamic merging process handles inputs sequentially. However, **it is straightforward to extend our approach to support merging in batches or groups**. We can achieve this by first obtaining router weights in batch, then grouping similar data items using the following strategy:
    1. Divide into Bins Based on Argmax Indices: First, we divide the data into several bins according to the arg-max indices of the router logits.
    2. Cluster Within Each Bin: Then, we cluster (by Kmeans) within each bin to group the logits (we set the group number to 20).
    3. Average Weights Within Each Group: Within each group, the router weights are averaged to obtain a merged model. Each group corresponds to one merged process, and the group size is typically larger than the batch size, making it very efficient.
        We have added a group-wise experiment on RoBERTa to illustrate this:

        | Method|  Avg. Normalized Score  | Time  |
        |-|-|-|
        | Task-Arithmetic | 67.80 |  4m52s  |
        | Twin-Merging | 96.14 |  9m31s  |
        | **Twin-Merging (group-wise)** | 92.02 |  5m14s  |

- We acknowledge the extra time cost due to routing and dynamical merging. However, as the inference process typically involves hundreds of forward passes (e.g., 300 tokens for summarization tasks), the additional computing is usually neglectable. Assuming context length $s$, task number $T$, layer number $m$, the introduced FLOPs ( Multiply–accumulate operation ) can be computed as $m(24sh^2 + 4bs^2h)$ for routing, $Tm(12h^2 + 9h)$ for merging (excluding norm parameter), while generating $L$ tokens typically requires $ \sum_{l=s}^{L} 24m(lh^2 + 4 bl^2h)$ FLOPs. **Given that $n \ll L$ and $s$ are typically truncated, the additional consumption is neglectable**.
We demonstrate the actual time cost in Table 7, which adds only 0.039 seconds per sample while bringing significant performance improvements.

- Moreover, our approach offers significant performance improvements with these additional computing resources.
As shown in Table 2 and the "More Baseline" section, we achieve an absolute normalized improvement of 28.34% for RoBERTa, 18.83% on ViT-B-32 compared to Task Arithmetic, 9.71% compared to Twin-Merging on Qwen-14B.
Traditional model merging methods often overlook the heterogeneous nature of test inputs, leading to substantial performance gaps.
Advanced merging techniques like AdaMerging and Surgery typically require costly training and searching processes, as demonstrated in the "More Baseline" section.
**In contrast, our method achieves superior performance to fine-tuned models with minimal cost and storage requirements (47m22s vs 215m01s, 5.0GB vs 32.4GB) due to dynamic merging and SVD techniques.**

- Furthermore, after merging, inference uses a single model per batch. This allows us to leverage optimizations like KV cache, Group Query Attention, efficient FFN/attention, and model compression techniques. Our method is also compatible with inference engines like FlashDecoding, DeepSpeed, and vLLM.

[1] AdaMerging: Adaptive Model Merging for Multi-Task Learning [ICLR24]

[2] Representation Surgery for Multi-Task Model Merging [ICML24]

---

### Decision · Program_Chairs · 2024-09-25

**Decision:**

Accept (poster)

**Comment:**

The work proposes Twin-Merging, a two-stage method for model merging: (1) modularizing knowledge into shared and exclusive components, with compression to reduce redundancy and enhance efficiency; (2) dynamically merging shared and task-specific knowledge based on the input. During testing, Twin-Merging dynamically merges shared and compressed specialized knowledge based on test inputs to form the final inference model. Compared to conventional merging methods, the new method better addresses adaptability to data heterogeneity and therefore further bridges the gap between merged models vs. fine tuned models.

The final recommendations from the reviewers are unanimously acceptance (5, 6, 6, 6).

The paper is well written.
The proposed method is technically sound.
The method is verified by quite extensive experiments.

The two major common concerns raised by most reviewers are more baselines and discussion/evaluation over inference efficiency. Both have been addressed in the rebuttal. Please incorporate the new results and discussions in the final version.